# Y chromosome introgression between deeply divergent primate species

Axel Jensen ⬤[1] ✉, Emma R. Horton[2], Junior Amboko[2], Stacy-Anne Parke[3,4], John A. Hart[5], Anthony J. Tosi[6], Katerina Guschanski ⬤[1,7,8] ✉ & Kate M. Detwiler ⬤[2,8] ✉

Hybridization and introgression are widespread in nature, with important implications for adaptation and speciation. Since heterogametic hybrids often have lower fitness than homogametic individuals, a phenomenon known as Haldane's rule, loci inherited strictly through the heterogametic sex rarely introgress. We focus on the Y-chromosomal history of guenons, African primates that hybridized extensively in the past. Although our inferences suggest that Haldane's rule generally applies, we uncover a Y chromosome introgression event between two species ca. six million years after their initial divergence. Using simulations, we show that selection likely drove the introgressing Y chromosome to fixation from a low initial frequency. We identify nonsynonymous substitutions on the novel Y chromosome as candidate targets of selection, and explore meiotic drive as an alternative mechanism. Our results provide a rare example of Y chromosome introgression, showing that the ability to produce fertile heterogametic hybrids likely persisted for six million years in guenons.

Interspecific hybridization is widespread in nature, and is an increasingly acknowledged evolutionary force[1]. However, when divergent lineages interbreed, the resulting hybrids often show a sex-specific reduction in fitness. This phenomenon was first described a century ago by Haldane, who stated that 'when in the offspring of two different animal races one sex is absent, rare, or sterile, that sex is the heterozygous sex'[2]. Haldane's rule has been confirmed by observations in nature and laboratory across diverse species with sex chromosomes, including animals and plants[3–5].

Several mechanisms have been suggested as causes of Haldane's rule[6], most of which predict that heterogametic hybrids will suffer greater consequences of genomic incompatibilities (Bateson-Dobzhansky-Muller incompatibilities, BDMI[7]). The dominance theory is most frequently invoked[5], stating that any BDMI involving the X/Z chromosome will be exposed to selection in heterogametic hybrid individuals, whereas homogametic hybrids only suffer the fitness costs of dominant incompatible alleles. A somewhat related explanation is the Faster-X/Z theory, suggesting that the X/Z tend to have accelerated evolutionary rates relative to the autosomes, which may reinforce the dominance effect. The faster male theory is also commonly proposed as a cause of Haldane's rule in male heterogametic species. It is based on the prediction that males are exposed to higher selection pressure than females (joint effects of natural and sexual selection), leading to an accelerated divergence in male reproductive traits. These examples are just a subset from an array of proposed causes of Haldane's rule, and it is likely that several mechanisms often interact.

As a consequence of Haldane's rule, gene flow between divergent lineages is more likely to occur through homogametic hybrid individuals, and loci that are strictly inherited through the heterogametic sex are unlikely to introgress[8–10]. Consequently, hybridization

[1]Department of Ecology and Genetics, Animal Ecology, Uppsala University, Uppsala, Sweden. [2]Department of Biological Sciences, Florida Atlantic University, Boca Raton, FL, USA. [3]Department of Anthropology, New York University, New York, NY, USA. [4]New York Consortium in Evolutionary Primatology, New York, NY, USA. [5]Lukuru Wildlife Research Foundation, Kinshasa, Democratic Republic of Congo. [6]Department of Anthropology and School of Biomedical Sciences, Kent State University, Kent, OH, USA. [7]School of Biological Sciences, Institute of Ecology and Evolution, University of Edinburgh, Edinburgh, UK. [8]These authors contributed equally: Katerina Guschanski, Kate M. Detwiler. ✉e-mail: axejen@gmail.com; katerina.guschanski@ed.ac.uk; kdetwile@fau.edu

frequently leads to phylogenetic discordance between autosomes and loci inherited only through the homogametic sex (like the mitochondrial genome [mtDNA] in mammals[11–13]), whereas the phylogeny of loci limited to the heterogametic sex (e.g. Y/W chromosomes) tend to agree with the autosomal tree[14–16]. Only a few cases of Y chromosome introgression have been reported in mammals[15,17–20], all between relatively close lineages (<3 million years divergent). In several of these examples, positive selection was proposed as a facilitator of the Y chromosome introgression[15,21], and specifically meiotic drive, realized through gene copy-number expansions on the sex chromosomes, was suggested in mice[22].

Here, we investigate the presence of Y chromosome introgression among lineages that experienced ample gene flow throughout their evolutionary history. Specifically, we focus on guenons, a group of African primates with more than 30 recognized species[23–25]. Guenons are renowned for their ability to hybridize[26–28], and interspecific gene flow was highly prevalent in their past[12,16,29]. Furthermore, male guenons typically disperse upon maturation, whereas females remain resident[30], providing opportunities for the Y chromosome to introgress across species boundaries. Thus, guenons are an ideal system to study the impact of Haldane's rule in the context of gene flow, along a speciation continuum. Although we find that Y chromosomal and autosomal phylogenies generally agree in guenons, in line with the expectations under Haldane's rule[5], we uncovered a Y chromosome introgression event between deeply divergent lineages, at a temporal scale that is unprecedented in mammals. Using simulations, we demonstrate that the Y chromosome was introgressed at a low initial frequency, and most likely driven to fixation by a non-neutral process.

## Results

### Sequencing and genotyping
We expanded a recently published dataset[12,31] by generating whole genome sequencing data from two previously unsequenced guenon species: *Cercopithecus denti* (one male, one female) and *C. wolfi* (two males, one female). We also sequenced one male each of *C. mitis* (ssp. *stuhlmanni*) and *C. hamlyni*. Together with other published guenon genomes[16,29] and two outgroup species from the sister tribe Papionini[31], our final data set contained 57 samples from 26 species (Supplementary Data 1). All genomes were sequenced to high coverage (≥19 x, Supplementary Data 1) using the Illumina platform. The reads were mapped and genotyped against the rhesus macaque (*Macaca mulatta*) reference genome (Mmul_10, GenBank: GCA_014858485.1), yielding on average 2.2 genotyped gigabases (Gb) per sample.

### Phylogenies from markers of different inheritance modes reveal a deep autosomal versus Y-chromosomal conflict in *Cercopithecus denti*
We inferred the guenon species tree with ASTRAL[32] from 5037 independent, autosomal gene trees (Fig. 1A). The species tree topology was in agreement with that reported by ref. 12, and all genera, species groups, and species, where more than a single sample was sequenced, were monophyletic with complete local posterior probability support (LPP = 1). As expected, the newly sequenced *Cercopithecus denti* and *C. wolfi* show a sister species relationship and cluster together with the phenotypically similar and geographically close *C. pogonias*[23,24], hereafter referred to as the eastern *mona* clade (Fig. 1A).

We also estimated the mitochondrial phylogeny using IQTree[33]. While *C. denti* and *C. wolfi* showed similar placements as in the species tree, the mitochondrial topology confirmed the previously described, extensive mito-nuclear discordances among guenons (Supplementary Fig. 1), which are consequences of rampant ancestral gene flow combined with incomplete lineage sorting (ILS)[12].

To infer the Y-chromosomal phylogeny, we constructed a maximum likelihood tree based on a Y chromosome alignment of all males in our dataset, using IQTree (Fig. 1A, hereafter referred to as the Y-

tree). In contrast to the mitochondrial phylogeny, the Y-tree was very similar to the species tree, as would be expected under Haldane's rule. *Cercopithecus denti*, however, stands out: In stark contrast to its position in the species tree as a *mona* group member, *C. denti* is nested within the *mitis* group on the Y-tree. The closest Y chromosome relative to *C. denti* is *C. mitis opisthostictus* (Fig. 1A), with a current distribution range south of *C. denti* (Fig. 2). The *mitis* group contains the taxonomically poorly resolved *C. mitis* and *C. albogularis* species, which form a paraphyletic clade in our analyses (Fig. 1A). For simplicity, we treat this clade as a single lineage, referred to as *C. mitis* hereafter, and use *C. m. opisthostictus* as the representative of this lineage in downstream analyses. Using MCMCTree[34], we estimated the autosomal divergence time between the *mitis* and *mona* groups to ca. 8 million years ago (Mya), and the Y-chromosomal divergence between *C. denti* and *C. mitis opisthostictus* to ca. 1.9 Mya (Fig. 1B, Supplementary Figs. 2–4). If introgression indeed caused this deep discordance between the species tree and the Y-tree, the Y chromosome introgression happened between lineages that diverged more than 6 million years earlier, according to our inferences.

Additional discordances between the species tree and the Y-tree were shallow, involving lineages that differ in position across a single speciation node (Fig. 1A). Although they may be the result of Y chromosome introgression, such patterns can be expected under ILS alone[12].

### Introgressive hybridization led to the fixation of a divergent Y chromosome
We sequenced the Y-linked *TSPY* locus from four additional *C. denti* males across the species' distribution range, all of which carried the *mitis*-like Y-chromosome (Supplementary Fig. 5, Supplementary Data 2). We therefore refer to the *mitis*-like Y chromosome as fixed in *C. denti* hereafter, although we acknowledge that a higher sample size is needed to confirm this. To test if the deep discordance between the Y-tree and the species tree was indeed caused by introgression rather than ILS, we compared pairwise nucleotide divergence ($d_{XY}$, average number of nucleotide differences per site) among *mona* and *mitis* group taxa on the autosomes and the Y chromosome, since introgression and ILS would generate distinct $d_{XY}$ patterns. For ILS to generate the discordant Y-tree placement of *C. denti*, the Y chromosome must have been polymorphic in the ancestor of the *mona/mitis/cephus* groups and remained unsorted along their descending branches (Fig. 3A). In this scenario, the Y chromosome of *C. denti, C. mitis, C. nictitans*, and the *cephus* group would all coalesce in their common ancestor, producing a greater (or similar) divergence on the Y chromosome compared to the autosomes. Under introgression, the divergence between *C. mitis* and *C. denti* should be much lower on the Y chromosome than on the autosomes (Fig. 3B), whereas the divergence between *C. denti* and its autosomal sister *C. wolfi* should be higher on the Y chromosome than on the autosomes. For taxa comparisons not involving *C. denti*, similar Y-chromosomal and autosomal $d_{XY}$ values are expected.

Nucleotide divergence between *C. mitis* and *C. denti* was on average 3.6 times greater on autosomal compared to Y-chromosomal loci (Fig. 3C), consistent with introgression of the Y chromosome from *C. mitis* into *C. denti* (Fig. 3B). Furthermore, the $d_{XY}$ estimates among *mitis* and *cephus* group species showed similar values for the Y chromosome and the autosomes (Fig. 3C), suggesting that sorting of the Y chromosome along the ancestral branches of these lineages was complete, hence opposing the expected ILS pattern.

### Low levels of autosomal allele sharing suggests that the Y chromosome introgressed at a low initial frequency
To test for autosomal gene flow corresponding to the Y chromosome introgression from *C. mitis* into *C. denti*, we estimated D-statistics[35] for different combinations of *mona* and *mitis* group taxa (Fig. 4). Using the

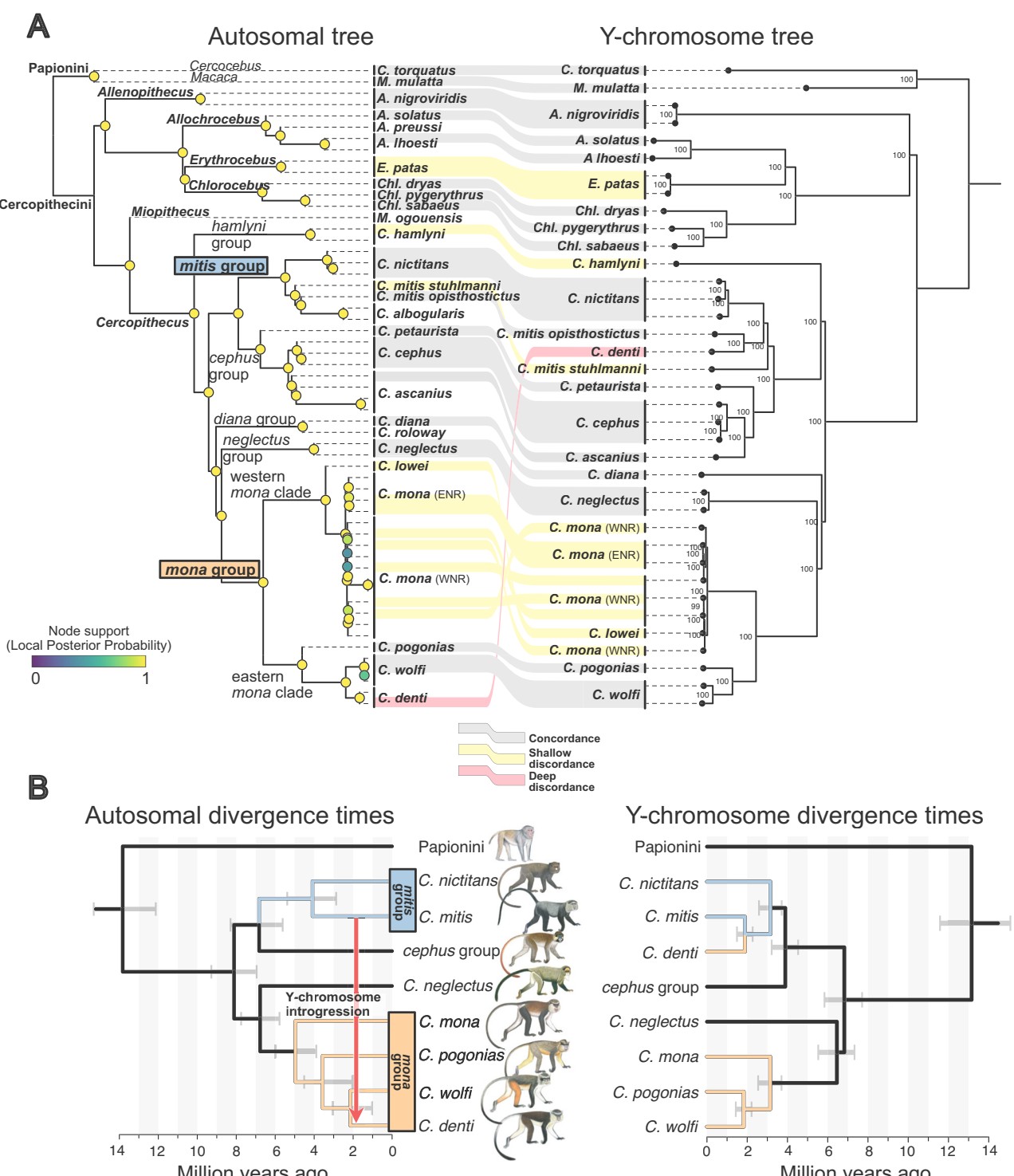

**Fig. 1 | Autosomal and Y-chromosomal phylogenies of guenons. A** ASTRAL coalescent-based tree constructed from autosomal data (left) and a maximum likelihood tree based on a Y chromosome alignment (right). ASTRAL local posterior probability support is indicated by colored circles on the autosomal tree nodes, and bootstrap node support (%) is indicated by labels in the Y chromosome tree. Connectors show the Y-chromosomal position of male samples relative to the autosomal tree: Concordant phylogenetic positions between the autosomal and the Y-chromosomal tree are shown by gray connections. Shallow discordances that can be explained by either ILS or introgression are shown in yellow, whereas the deep discordance for the placement of *C. denti* that cannot be explained by ILS is shown in red. Following ref. 29, we separate *C. mona* in two populations, as indicated by labels in parentheses (ENR and WNR, east/west of the Niger river, respectively). **B** MCMCTree divergence date estimates among focal lineages, based on autosomal (left) and Y-chromosomal (right) data, with error bars on nodes representing 95% highest posterior density intervals. The arrow in the autosomal tree depicts the inferred Y chromosome introgression. Primate illustrations copyright 2013 Stephen D. Nash/IUCN SSC Primate Specialist Group. Used with permission. Source data are provided as a Source Data file.

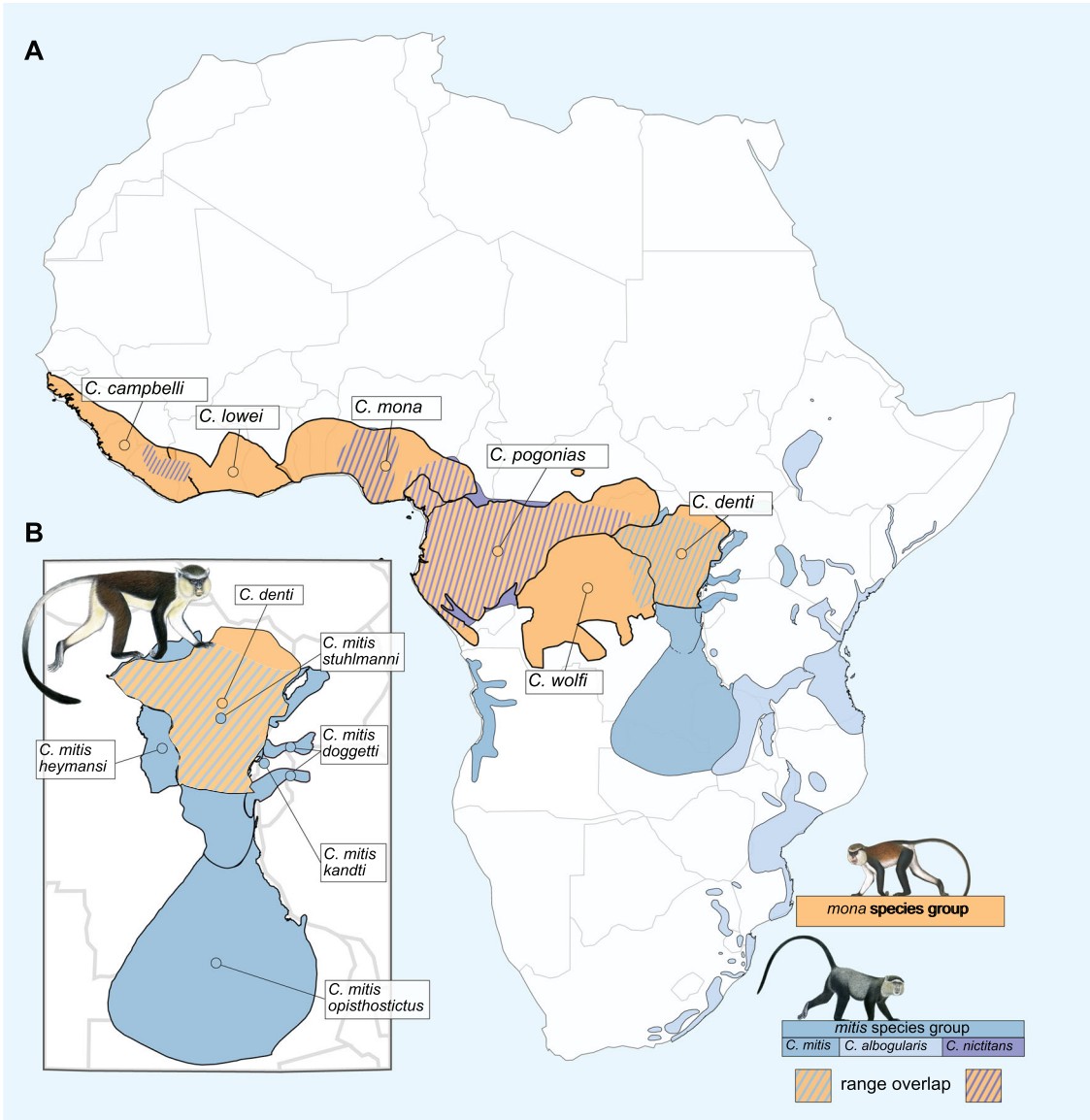

**Fig. 2 | Distribution ranges of the *mona* and *mitis* group lineages. A** Distribution ranges of recognized *mona* and *mitis* group species[24]. **B** Zoom-in around the distribution range of *C. denti*, showing its overlap with, and close proximity to, multiple *C. mitis* subspecies. Primate illustrations copyright 2013 Stephen D. Nash/IUCN SSC Primate Specialist Group. Used with permission.

rhesus macaque as outgroup, we tested for excess autosomal allele sharing (indicative of gene flow) between *C. mitis/nictitans* and *C. denti*, compared to *C. mona, C. pogonias* and *C. wolfi*, iterating through all combinations of samples of the respective species. We detected a strong excess of allele sharing between all *mitis* group taxa and *C. denti* compared to *C. mona* (Fig. 4A, Supplementary Data 3). This is consistent with the previously reported allele sharing between the *mitis* group and the eastern *mona* clade[12]. However, ref. 12 showed that this signal was mainly driven by shared ancestry between the *mitis* and *cephus* groups, as it disappeared in *mitis* but not in *cephus* when considering only variants private to either lineage. Thus, since this gene flow event predominantly occurred from *C. cephus* rather than from *C. mitis*, and into the common ancestor of *C. denti, C. wolfi* and *C. pogonias*, it is unlikely to have introduced a *mitis*-like Y chromosome only into *C. denti*.

Relative to its closest sister species (*C. pogonias* and *C. wolfi*), we found only weak signals of excess allele sharing between *C. denti* and *C. mitis*, with only five out of 56 tests producing significant D-values (*Z*-score > 3, Fig. 4B, C, Supplementary Data 3).

Furthermore, we found no excess allele sharing between *C. denti* and specifically its closest Y-chromosomal relative *C. m. opisthostictus*. Overall, these analyses provide weak support for autosomal gene flow from *C. mitis* into *C. denti* after the split from *C. pogonias* and *C. wolfi*. This suggests that the effective migration rate was low, and the *mitis*-like Y chromosome of *C. denti* was likely introduced at a very low initial frequency.

As expected from the weak signal of autosomal gene flow from *C. mitis* into *C. denti*, a sliding window topology analysis showed that the Y-tree topology was rare on the autosomes and absent on the X chromosome (Supplementary Fig. 6). We found only 11 autosomal regions of 10 kb where *C. denti* was nested within the *mitis* group as sister to *C. mitis*. Pairwise $d_{XY}$ comparisons in these regions (similar to those performed for the Y chromosome in Fig. 3) show a pattern more compatible with introgression than ILS. The $f_d$ statistic, which quantifies excess allele sharing in genomic windows, was also increased in these regions relative to the genome-wide average, supporting introgression (Supplementary Fig. 6). Four protein coding genes, with functions in, e.g., coordination and muscle strength, kidney function

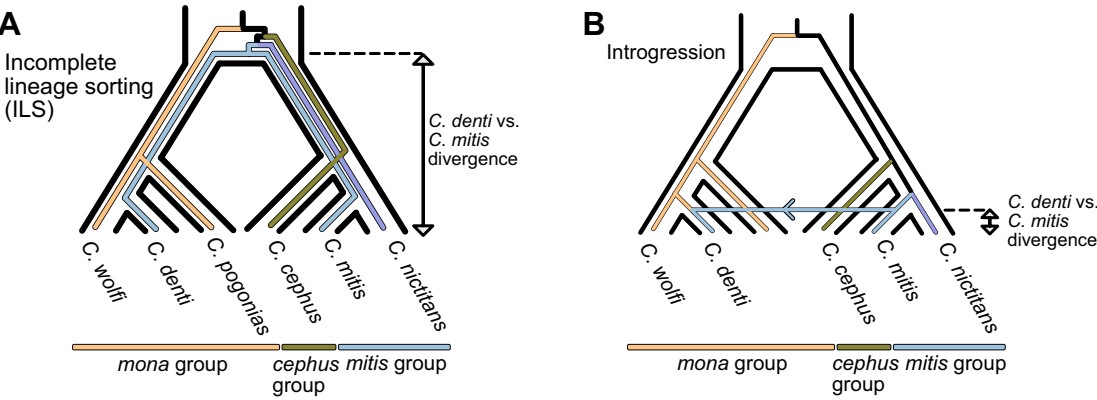

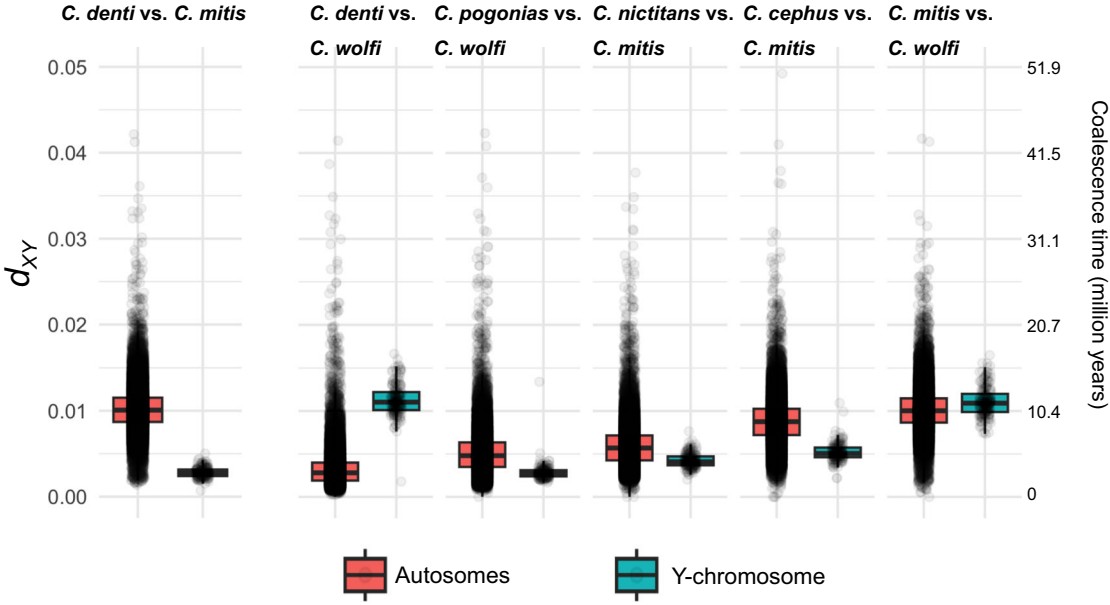

**Fig. 3 | Nucleotide divergence patterns support Y chromosome introgression over ILS in *C. denti.*** Schematics of Y chromosome sorting in the *mona* and *mitis/cephus* group lineages under ILS (**A**) and introgression (**B**). Colored lines represent Y lineages, whereas black outlines show the species tree relationships. The arrow in B corresponds to Y-chromosomal introgression from *C. mitis* into *C. denti.* Double-headed arrows on the right in **A** and **B** reflect expected nucleotide divergence ($d_{XY}$) for the Y chromosome between *C. denti* and *C. mitis.* **C** Nucleotide divergence ($d_{XY}$) in 50 Kb autosomal ($n = 52{,}826$) and Y-chromosomal ($n = 203$) windows between taxa present in (**A**) and (**B**), supporting introgression of the Y chromosome from *C. mitis* into *C. denti.* Second axis on the right shows the nucleotide divergence scaled to coalescence time. Boxplot elements: center line, median; hinges, first/third quartile; whiskers, observations in 1.5× interquartile range. Source data are provided as a Source Data file.

and body size (*BRINP3*, *RPL18*, *RNF170* and *SLC7A9*, ref. 36) overlapped the putatively introgressed autosomal regions.

Although most of the Y chromosome is haploid and non-recombining, the primate sex chromosomes typically contain at least one pseudo-autosomal region (PAR), where X and Y recombine[37]. We identified one PAR at the start of the reference macaque X chromosome (PAR1, ca. 2.36 Mb in length) using a mapping coverage based approach (Supplementary Fig. 7). No PAR-like regions were assembled/present on the reference macaque Y-chromosome (Supplementary Fig. 7). There was no signal of increased introgression from *C. mitis* into *C. denti* on the PAR relative to the non-PAR X or autosomes (Supplementary Fig. 8A–C). The PAR recombines at a rate of up to 20 times higher than the autosomes[38,39], which likely rapidly broke down the linkage between the introgressing Y and PAR. Indeed, using simulations, we found that introgression on PAR was similar to that of the autosomes already at a tenfold increase in recombination rate relative to the genome-wide average, despite Y chromosome introgression (Supplementary Fig. 8D). In line with reports of higher substitution

rates in the primate PAR1[40], we found greater divergence on the PAR, relative to the non-PAR X and autosomes, between both *C. denti* and *C. mitis* (Supplementary Fig. 8B), and *C. denti* and *C. wolfi* (Supplementary Fig. 8C).

**Genetic drift is an unlikely driver of the Y chromosome fixation**
After establishing that the *mitis*-like Y chromosome must have introgressed at a low initial frequency, we used simulations to investigate if its fixation in *C. denti* could be explained by drift alone or if it was facilitated by a non-neutral process. As a first step, we explored plausible proportions of migration from *C. mitis* into *C. denti* given our empirical D-statistic estimates. We used the multi-species-coalescence-with-introgression (MSci) model implemented in BPP[41] to infer the demographic history of the focal lineages as a basis for subsequent simulations. We included three gene flow events in our model (Fig. 5): (1) from the *mitis* group ancestor into the ancestor of *C. pogonias*, *C. wolfi* and *C. denti*, (2) from the *C. cephus* lineage to the ancestor of *C. pogonias*, *C denti* and *C. wolfi* (as inferred in ref. 12), and (3) from *C.*

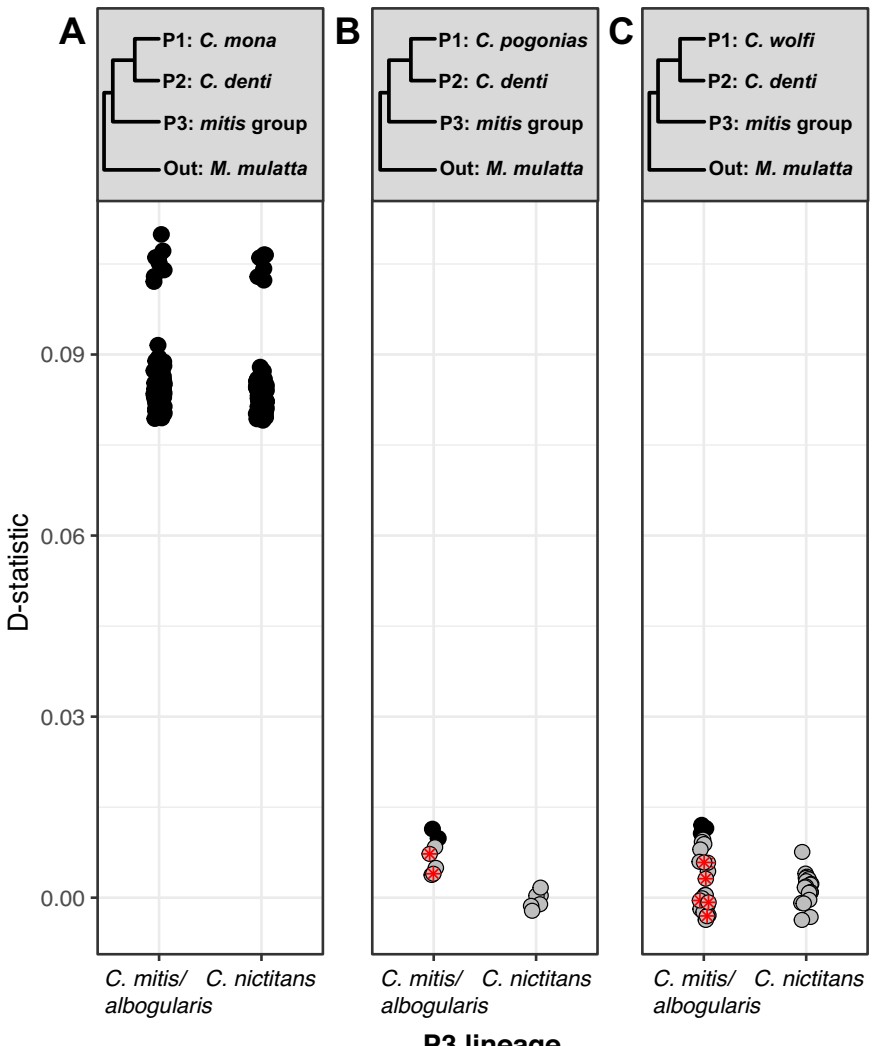

**Fig. 4 | Negligible autosomal introgression from *C. mitis* into *C. denti*.** Estimated excess allele sharing between *C. denti* and *mitis* group taxa relative to *C. mona* (**A**), *C. pogonias* (**B**), and *C. wolfi* (**C**). Filled circles show significant D-statistics (*Z*-score > 3). *Z*-scores were obtained using a block-jackknife standard error of the D-statistic (20 blocks per chromosome, ref. 72). All sample combinations consistent with the trees indicated in panel headers were tested, and each circle represent one such test. Circles with red asterisks in (**B**) and (**C**) show tests where the *C. m. opisthostictus* was used as P3. Source data are provided as a Source Data file.

*mitis* into *C. denti* (corresponding to the Y chromosome introgression). Although there is no direct support for the gene flow event 1[12], it is possible that it could partially be masked by extensive gene flow from the *cephus* group into the *mona* group (event 2). We therefore conservatively included this event since it could be a source of the Y chromosome introgression (together with ILS) into *C. denti*, and have an effect on the observed D-statistics.

Two independent runs of BPP-MSci with the same parameters converged on highly similar estimates (Supplementary Fig. 9). Scaling the divergence times using a generation time of 10 years and a mutation rate of 4.82e-9 substitutions per bp per generation[31], resulted in estimates similar to the MCMCTree analysis (Figs. 5, 1B), supporting that ca. 6 million years of divergence preceded the Y chromosome introgression from *C. mitis* into *C. denti*.

Ancestral effective population size estimates were generally large (>100,000, Fig. 5), in line with the reported high genetic diversity in guenons[12,31]. The most pronounced gene flow event was, as expected,

from *C. cephus* into the eastern *mona* clade (event 2 in Fig. 5, migration rate [phi] = 8.5%, time ~2.7 MYA), followed by event 1 (migration rate ca. 2.1% at ~3.1 MYA.). In line with our D-statistics result, the migration rate from *C. mitis* into *C. denti* (event 3 in Fig. 5) was low (phi = 0.2% ~0.3 MYA). Switching the order of the gene flow events 1 and 2 resulted in similar estimates overall, but lower migration proportions in event 1 (phi = 0.8%, Supplementary Fig. 10).

Using the estimated demographic parameters (Fig. 5), we next performed simulations in msprime[42] to infer an autosomal migration rate from *C. mitis* into *C. denti* that is compatible with our empirical D-statistic estimates. We conservatively included both ancestral gene flow events 1 and 2. We tested a range of migration proportions from *C. mitis* into *C. denti* (0–1%, with a stepwise increase of 0.05%) and calculated D-statistics between *C. denti* and *C. mitis* compared to *C. wolfi*. The gene flow was set to occur shortly (10,000 generations) after the split between *C. denti* and *C. wolfi* to allow drift to act on the introgressing loci.

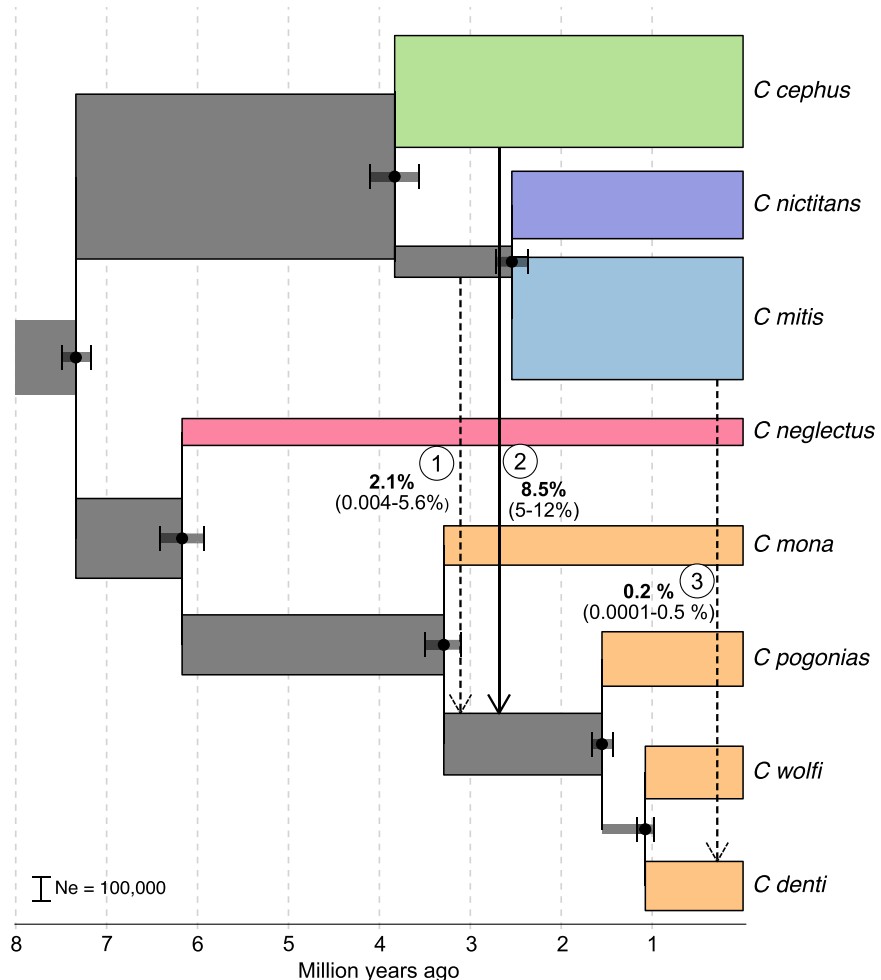

**Fig. 5 | Demographic history of focal lineages as estimated with BPP-MSci.** Gene flow events included in the demographic model are shown as arrows, signifying their directionality. Labels next to the arrows show the obtained means and 95% highest posterior density intervals of estimated migration rate for each event. Dashed arrows indicate gene flow events with low support from D-statistics estimates (Fig. 4, ref. 12). Widths of the branches correspond to inferred effective population sizes (Ne), as indicated by the scale bar. Source data are provided as a Source Data file.

Our simulations showed that migration rates of >0.4% consistently produce significantly positive D-statistics (Z-score > 3, Fig. 6A). This migration proportion also generated consistently greater D values than our highest empirically observed D-statistic. For computational reasons, the simulated genome length was 100 Mb, i.e. considerably smaller than the primate genome (corresponding to only ~3.5%). As the sensitivity of D-statistics increases with the number of loci[43], it is therefore highly unlikely that the effective migration rate from *C. mitis* into *C. denti* (that resulted in Y chromosome introgression) exceeded 0.4%.

Under the conservative assumption of exclusively male dispersal, the effective Y-chromosomal migration rate is expected to be twice that of the autosomal migration rate. As the fixation probability of a novel allele equals its initial frequency[44], our estimates suggest that the maximum probability of the introgressing *mitis*-like Y chromosome drifting to fixation in *C. denti* is 0.8% (2 * 0.4%). However, this estimate does not consider the possibility that the Y chromosome introgressed from the ancestral *mitis* group lineage into the eastern *mona* clade, and generated the *C. denti/C. mitis* sister relationship through ILS (event 1, Fig. 5). Therefore, we also performed simulations that estimate the frequency of *C. denti/C. mitis* monophyly with all gene flow events included, which produced similar results (Supplementary Fig. 11).

## Moderate positive selection is capable of driving the Y chromosome fixation in *C. denti*

Having established that drift is an unlikely cause of the Y chromosome fixation, we investigated how different strengths of selection affect the fixation probability under plausible migration rates. We used SLiM[45] to simulate a population of 200,000 individuals for 100,000 generations (mimicking the *C. denti* lineage based on our BPP analysis, Fig. 5) and traced the frequency of a novel Y-chromosomal allele introduced at different initial frequencies (i.e. introgression proportions), with varying selection coefficients (s).

As expected, the novel Y chromosome was highly unlikely to fix without selection: Drift alone drove the novel allele to fixation a single time across 600 simulations (with initial Y chromosome frequencies ranging between 0.1 and 1%), and it remained segregating after 100,000 generations in four additional replicates (Fig. 6C). However, with a moderate selection coefficient of s = 0.001, the fixation rate of the novel Y chromosome was 62% already at an initial frequency of 0.4% (equivalent to the estimated migration rate from BPP-MSci), and 78% when the initial frequency was 0.8% (equivalent to the upper bound of migration based on empirical D-statistics and simulations). Hence, these simulations together with theoretical expectations suggest that the introgressing Y chromosome likely had a selective advantage over the ancestral Y, allowing it to reach fixation from a low initial frequency.

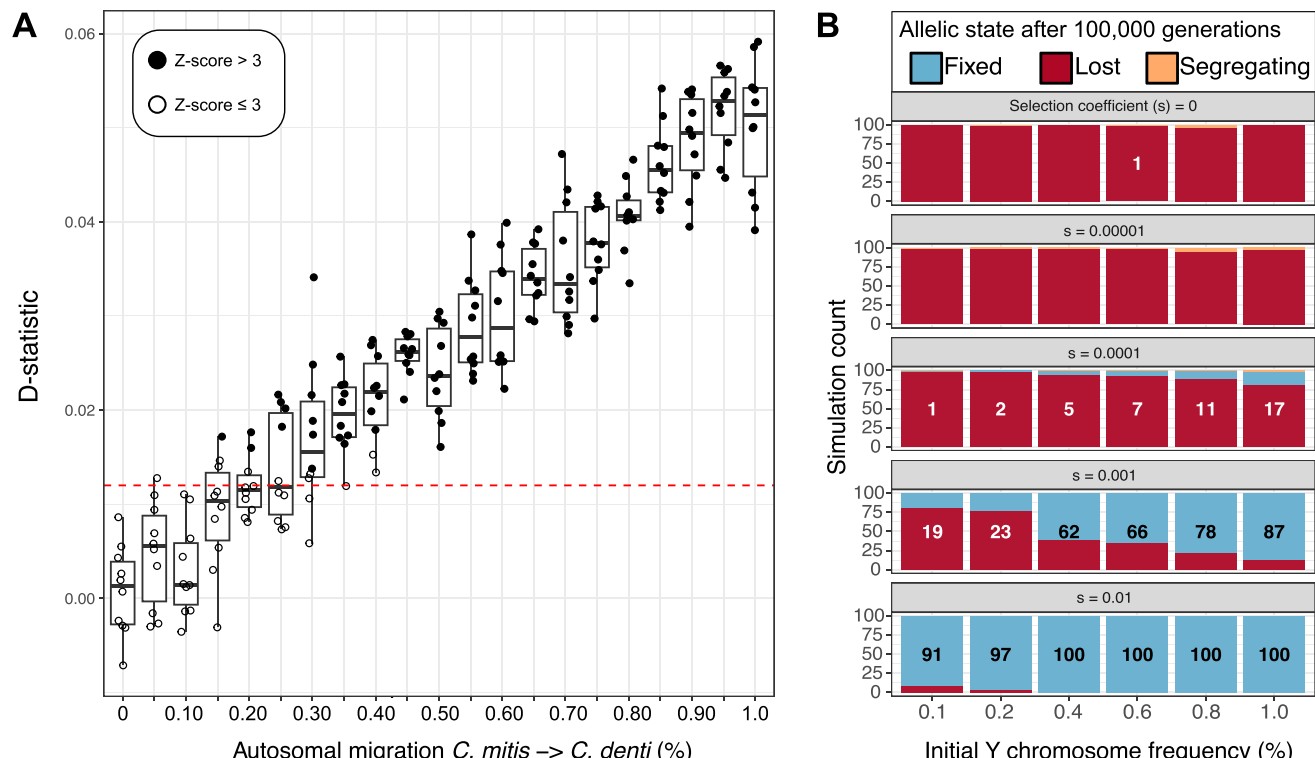

**Fig. 6 | Simulated D-statistics and Y chromosome fixation rates. A** D-statistics (P1 = *C. wolfi*, P2 = *C. denti*, P3 = *C. mitis*, Outgroup = *M. mulatta*) resulting from simulated evolutionary histories with varying amounts of migration from *C. mitis* into *C. denti*. Each circle represents the estimated D-statistics in one simulation, with increasing gene flow proportion along the *X*-axis. Ten simulations were performed for each migration rate, and *Z*-scores were obtained from the block-jackknife standard error of the D-statistic, using 20 blocks for each simulation. Filled circles represent significantly positive D-statistic tests (*Z*-score > 3), and the horizontal red dashed line depicts the greatest observed D-value from empirical data. Boxplot elements: center line, median; hinges, first/third quartile; whiskers, observations in 1.5× interquartile range. **B** Simulated Y chromosome fixation rates under different initial allele frequencies (equivalent to incoming Y-chromosomal migration, *X*-axis) and selection coefficients (panels), using forward simulations in SLiM. Numbers on bars state the count of simulations resulting in fixation of the introgressing Y chromosome, out of a total of 100 replicate simulations. Source data are provided as a Source Data file.

## Possible targets of selection on the introgressing Y chromosome

We explored possible targets for positive selection on the introgressing Y chromosome by identifying Y-linked genes with fixed amino acid substitutions in the *mitis* group and *C. denti* compared to *C. pogonias* and *C. wolfi*. Of the 41 annotated Y-chromosomal genes on the macaque reference genome, only 28 contained any sites that passed filtration in our study genomes, as a consequence of the highly repetitive nature of the Y chromosome (Supplementary Data 4, see "Methods"). After removing genes with premature stop codons in any of the focal species, we identified 13 genes with at least one fixed amino acid difference in *C. mitis* and *C. denti* compared to *C. wolfi* and *C. pogonias* (Supplementary Data 4). The highest numbers of substitutions were found in the genes *KDM5D* and *USP9Y*, with five and four differences, respectively.

We also searched for evidence of adaptive evolution in these thirteen genes with two different approaches. First, we used codeml implemented in the PAML package[34] to specifically test whether any sites show signatures of adaptive evolution along the ancestral *C. mitis/C. denti* Y chromosome branch. No such sites were identified (likelihood ratio test [LRT], *p* > 0.25). Second, we used the MEME method implemented in the HyPhy suite[46] to identify sites that show signals of episodic diversifying selection. Four such sites were identified (LRT, *p* = 0.080–0.087, Supplementary Data 5). Three of these, located in *DAZA*, *ZFY* and *KDM5D*, were differentially fixed between the eastern *mona* clade and *C. denti/C. mitis*, highlighting these genes as candidate targets for selection.

An alternative to the classical scenario of adaptive introgression is meiotic drive orchestrated by selfish genetic elements. One reported drive mechanism for the Y chromosome is through the expansion of gene copies, which can increase the proportion of Y-bearing sperm relative to X-bearing sperm[22,47]. This may then be compensated for by copy-number expansions on the X-chromosome, leading to an arms race between the sex chromosomes. During hybridization, the lack of compensatory expansions on the X-chromosome in a naive genome may allow for rapid drive of the introgressing Y chromosome. This scenario was implicated in a Y chromosome introgression between mouse subspecies[18], and selfish arms-races between the X and Y has been suggested to be a widespread phenomenon in mammals[47]. To explore this possibility, we investigated the mapped read coverage in *C. denti* and other guenon species after mapping to the rhesus macaque Y chromosome, as this could indicate a copy number expansion. We identified a region of ca. 1.35 Mb with 2–7 times higher normalized coverage in *C. denti* compared to *C. wolfi* and higher coverage in *C. denti* than in any other *mona* group species (Supplementary Fig. 12, 13). This region contains two genes from the *CDY* gene family (*CDY* and *LOC106995435*). However, no evidence of higher copy number in this region was present in *C. m. opisthostictus*, the Y-chromosomal sister of *C. denti*, contradicting that this region constitutes an introgressed drive element. Furthermore, we found no evidence of compensatory copy-number expansions on the *C. denti* X-chromosome relative to *C. wolfi* (Supplementary Fig. 14). The signature of a copy-number expansion in this 1.35 Mb Y-chromosomal region was also present in other guenon species (Supplementary Fig. 15). While it is possible that this or some other undetected Y chromosome region has been implicated in meiotic drive in *C. denti*, our results are inconclusive. Resolving the ampliconic structure of the Y chromosome, which is possible

only through the use of long-read sequencing data, will provide a better understanding of these processes[48].

## Discussion

In this study, we investigate the prevalence of Y chromosome introgression in a species group that experienced extensive hybridization throughout their evolutionary history. Although the Y-chromosomal phylogeny was generally in agreement with the species tree, we uncover the most distant Y chromosome introgression reported in mammals to date. We explore the evolutionary processes behind this event that occurred between primate lineages that, according to our inferences, diverged from each other more than six million years prior to the introgression (corresponding to ca. 600,000 generations of independent evolution). As illustrated by the near-absence of autosomal introgression, the Y chromosome swept to fixation in *C. denti* from a very low initial frequency (≤0.8%). While this is highly unlikely under neutral drift alone, we demonstrate that moderate positive selection is sufficient to achieve fixation of the introgressing Y chromosome, even at very low levels of introgression.

The evolutionary history of guenons is characterized by rampant ancestral hybridization, involving deeply divergent lineages that experienced >5 million years of independent evolution and differ in chromosome numbers[12]. At least five introgression events of the maternally inherited mitochondrial genome (mtDNA) occurred across guenon species (Supplementary Fig. 1)[12,16,29], whereas the phylogeny of the paternally inherited Y chromosome is generally consistent with the species tree (Fig. 1A). Guenons show strong male-biased dispersal, which should create many opportunities for the Y chromosome to introgress across species boundaries.

Indeed, we observe a number of shallow discordances between the Y-tree and the species tree (Fig. 1A). Although ILS may be an important mechanism in generating such discordances, it is likely that some of them are a consequence of male-biased gene flow. For example, the closely related *Cercopithecus lowei* and the two populations of *C. mona* separated by the Niger river (*C. mona* WNR/ENR in Fig. 1A; ref. 29) form three well-defined clades in both the mitochondrial and autosomal phylogenies (Fig. 1A, S1). In the Y-tree they are intermixed, suggesting that male migration continued after initial population divergence. Similarly, *C. hamlyni*, another species involved in a shallow species tree vs. Y-tree discordance (Fig. 1A), has experienced gene flow with the ancestor of the *mitis* species group[12]. It is possible that *C. hamlyni* received gene flow already from the ancestor of the *mitis* and *cephus* species groups, making introgression a plausible explanation of its Y-tree placement. Finally, *Erythrocebus patas* shows variation in phylogenetic placement using different marker combinations and approaches, likely as a result of rapid speciation and pronounced ILS[12,31]. Even if Y chromosome introgression indeed caused these shallow discrepancies, none of the involved lineages would have been more than ca. 3 million years divergent at the time of introgression (e.g., *C. lowei* vs. *C. mona*). Although Y chromosome introgression is rare already on such time scales, comparable events have been reported previously. For instance, two species of macaques (*Macaca*) experienced Y chromosomal introgression ca. 2–3 million years after they diverged[19,49]. Y chromosome introgression has also been reported among baboons[17], canids[20], and mice[18], all involving lineages less than 1 million years divergent.

The prevalence of shallow Y-tree vs. species tree discordances confirms that the rampant gene flow experienced by guenons throughout their evolutionary history creates opportunities for Y-chromosomal introgression on a level rarely reported in other taxa. In contrast to the frequent mtDNA transfers, however, introgression events of more distant Y chromosomes were largely absent. This strongly suggests that male hybrids are affected by genomic incompatibilities to a greater degree than females, in line with Haldane's rule. In this context, the Y chromosome introgression into *C. denti* from the deeply divergent *C. mitis* constitutes a remarkable exception. For comparison, ref. 50 explored sequence divergence in the mitochondrial cytochrome *b* gene (*CYTB*) as a predictor of Haldane's rule, and found that fertile heterogametic hybrids were completely absent at divergences higher than 8%. Based on the contemporary *CYTB* divergence between *C. denti* and *C. mitis* of ~13–14%, these lineages would have had a divergence of >10% at the time of Y chromosome introgression (assuming a constant substitution rate). For context, this is comparable to the contemporary *CYTB* divergence between humans and chimpanzees (~10.8–11%, ref. 50), and further highlights the uniqueness of this introgression event.

Although backcrossing from heterogametic F1 hybrids is arguably the most parsimonious mechanism of the Y chromosome introgression in *C. denti*, another possibility is that the Y chromosome introgressed via an already admixed population. This could alleviate the negative selection against heterogametic hybrids to some extent, allowing the Y chromosome to introgress through crossings between female *C. denti* x *C. mitis* hybrids and male *C. mitis*, even in the absence of fertile male F1 hybrids. Nevertheless, a fertile male offspring produced by a male and a female that were predominantly *C. mitis* and *C. denti*, respectively, is still required for the Y chromosome to introgress into *C. denti*. Furthermore, the scarcity of Y/W chromosome introgression examples, despite many instances of hybridization and numerous reports of admixed populations, suggests that even introgression via an admixed population is rare[4,5], and/or that the selection against heterogametic hybrids remains strong at various levels of admixture.

Our analyses strongly suggest that a non-neutral process was involved in the fixation of the introgressing Y chromosome. As potential targets of positive selection, we identified several genes with fixed non-synonymous substitutions between the introgressed and ancestral Y chromosomes. The largest number of amino acid substitutions were found in *KDM5D* and *USP9Y* which, as most Y-encoded genes, have important functions in spermatogenesis[51,52]. Mutations or deletions of these genes have been reported to reduce male fertility in humans[53]. Sperm competition is an important selective force in many primates[54–56] and likely also in guenons, where influx of males into social groups during the breeding season has been reported[30]. Therefore, the two detected genes are plausible subjects for adaptive introgression. In line with this, our results also suggest that one of the fixed sites in *KDM5D* evolved under episodic diversifying selection.

Another possible mechanism behind the Y chromosome fixation is meiotic drive. Selfish genetic elements on the sex-chromosomes may act to increase the fertilization success of X or Y-bearing sperm, leading to an arms race between X and Y-linked genes[22,47]. Meiotic drive was implicated in the asymmetric Y chromosome introgression in a hybrid zone of house mouse subspecies (*Mus musculus* ssp.)[22], driven by a copy number expansion of the Y-linked *SLY* gene. If not compensated for by copy number increase of the homologous X-linked *SLX* gene, the *SLY* expansion leads to an excess of Y-bearing sperm and a male-biased sex-ratio. Although *SLY/SLX* are specific to the mouse/rat lineage, similar cases of co-amplification of X/Y-linked genes have been described in other mammals[47]. We detected a Y-chromosomal region showing signatures of extensive copy number variation among guenons, containing two genes from the *CDY* gene family. A rapid copy-number expansion in *CDY* genes was also reported in orangutan (genus *Pongo*) relative to other great apes, albeit without any apparent increase in gene expression[57]. Although the *C. denti* Y chromosome appeared to have higher coverage in this region compared to the other *mona* group lineages, the closest *C. mitis* Y chromosome does not show such a signature, which makes it unlikely that this region constitutes an introgressing drive element. We are currently lacking data on much of the *mitis* species group variation, however, and it is possible that the actual source of the Y chromosome is an unsampled relative of *C. m. opisthostictus*. If so, it is possible that the copy number

expansion occurred in this unrepresented lineage, as copy number expansions in this region were highly variable and species-specific (Supplementary Figs. 13, 15). Therefore, we cannot exclude meiotic drive as the underlying mechanism of the Y chromosome fixation at this point. The guenon Y chromosome thus provides an intriguing subject for future studies, and long read sequencing and testis expression data may help elucidate the mechanisms behind this introgression event.

Whichever mechanism is at play, our estimates of the selection coefficient needed to achieve Y chromosome fixation are likely underestimated. In our simulations, the introgressing Y chromosome is modeled as a neutral locus, whereas the underlying mechanism behind Haldane's rule results in selection against the foreign Y chromosome. Therefore, the selective advantage of the introgressing Y must have been strong enough to achieve fixation in the presence of negative selection. In addition, we assumed exclusively male migration in our simulations, whereas it is likely that some proportion of migrants were females. As a result, the initial frequency of the introgressing Y chromosome was likely lower than considered here, requiring stronger selection to drive it to fixation.

Interspecific gene flow between divergent lineages is a well-documented phenomenon[12,26,58,59]. As a consequence of Haldane's rule, the backcrossing required for introgression to occur is expected to be mediated mainly through hybrids of the homogametic sex (females in mammals), since the heterogametic hybrids are typically less fit. Hence, Y chromosome introgressions are rare in mammals, and if present, typically occur between closely related lineages that have accumulated few genomic differences. This study thus provides a notable exception, demonstrating Y chromosome introgression from *C. mitis* into *C. denti*, lineages that diverged more than six million years prior. Although the exact mechanisms that facilitated the introgression and enabled the fixation of this Y chromosome remain a mystery at this stage, we propose that selection either on coding genes or meiotic drive elements must have been strong enough to overcome the negative selection against heterogametic hybrids.

## Methods

### Sampling, DNA extraction and sequencing

Non-invasive tissue samples of *C. mitis* ($n = 1$), *C. hamlyni* ($n = 1$), *C. wolfi* ($n = 3$) and *C. denti* ($n = 5$) were collected opportunistically from deceased individuals in the Democratic Republic of Congo. We collected an additional non-invasive fecal sample of a male *C. denti* from Nyungwe National Park, Rwanda. Samples were stored in RNAlater or 94% ethanol. Field work and sample collection was performed under oversight of the TL2 Project of the Lukuru Wildlife Research Foundation, the Congolese Institute for Nature Conservation (permits ICCN/MB/DT/DG/2008-01188, ICCN/DG/ADG/KBY/2009-0660, ICCN/MB/DT/DG/2008-01188), and the Rwandan Development Board (formerly ORTPN and PCFN) (Supplementary Data 1). Sample export and import followed the CDC and U.S. Fish and Wildlife Service protocols, in compliance with CITES export regulations (CITES IDs 6718, 6719 and 3245). Total genomic DNA was extracted in the Primatology Lab at Florida Atlantic University (Institutional Biosafety Committee #2012-144, #2016-246) using the DNeasy Blood & Tissue Kit (Qiagen 69504; Germantown, MD) for the samples used for whole genome sequencing, and QiaAMP DNA Stool Mini Kit (Qiagen 51504; Germantown, MD) for the TSPY amplification (see below), following manufacturer protocols (Supplementary Data 1). For whole genome sequencing, DNA extracts were sent to the Science for Life Laboratory at Uppsala University, Sweden, where library preparation and sequencing was performed by the SNP&SEQ Platform using the TruSeq PCRfree DNA library preparation kit (Illumina Inc.). The libraries were sequenced on the NovaSeq 6000 platform, aiming at a sequencing depth of ca. 30 X. In addition, we obtained published medium to high coverage whole genome sequences from 50 additional individuals of 24 species,

including two outgroup species from the sister tribe Papionini, adding up to a total of 57 genomes from 26 species[12,16,29,31] (Supplementary Data 1).

### Mapping and variant calling

We followed the Genome Analysis Toolkit best practices workflow to process the data[60]. Briefly, we added read group information and marked adapters with Picard/2.23.4 and aligned the processed reads to the rhesus macaque *Macaca mulatta* reference genome (Mmul_10, GCF_003339765.1) using the mem algorithm in bwa/0.7.17[61]. The mapped BAM-files were sorted and deduplicated using Picard, and mapping quality and depth assessed with QualiMap/2.2.1[62]. Next, we used GATK/4.2 to first call genotype per samples with HaplotypeCaller in GVCF mode, which were then combined with CombineGVCFs and jointly genotyped with GenotypeGVCFs, set to output also invariant sites. Indels were excluded, and single nucleotide variants were filtered using VariantFiltration in GATK/4.2 following the recommended exclusion criteria (QD < 2.0, QUAL < 30.0, SOR > 3.0, FS > 60.0, MQ < 40.0, MQRankSum < −12.5, ReadPosRankSum < −8.0). Additionally, we used custom Python scripts to mask heterozygous sites with minor allele read support <0.25, and sites with less than half or more than twice the genome-wide average read depth for each sample (for the X and Y, we used the chromosome-wide average when calculating these cutoffs). Repetitive regions were identified and excluded following the SNPable regions pipeline (https://lh3lh3.users.sourceforge.net/snpable.shtml). Last, Y-chromosomal sites where any sample was called as heterozygous were removed.

### Phylogenetic analyses and divergence date estimates

We used ASTRAL/5.7.4[32] to infer the autosomal phylogeny for all species under the multispecies coalescent model. ASTRAL takes independent gene trees as input, for which purpose we sampled a 25 Kb alignment every 500 Kb and constructed a maximum likelihood tree with IQTREE/2.2.2.6[33] with 1,000 rapid bootstraps, using the GTR model. To infer the mitochondrial phylogeny, we assembled and annotated the mitochondrial genomes (mtDNA) of all samples with MitoFinder/1.4.1[63], after trimming the raw reads with TRIMMOMATIC/0.39[64], using a published *Chlorocebus sabaeus* mtDNA as reference (NC_008066.1). Each mtDNA genome was then divided into 42 partitions: 1st, 2nd and 3rd codon positions of 13 protein coding genes, two rRNA and 22 concatenated tRNA, which were individually aligned using MAFFT/7.407[65]. The best partitioning scheme and model was evaluated using the modeltest implemented in IQTREE, which was then also used to construct a maximum likelihood tree with 1000 rapid bootstraps. The Y-chromosomal phylogeny was constructed by first converting the filtered genotypes called against the rhesus macaque reference to a chromosome-wide alignment in fasta format using a custom Python script. After removing sites with more than 10% missing data, the final alignment consisted of 414,466 bp. A maximum likelihood tree was constructed in IQTREE using the GTR + F model of substitutions, performing 1,000 rapid bootstraps.

We estimated the divergence dates on the autosomes and Y chromosome separately using MCMCTree as implemented in PAML/4.9j[34]. We used a single sample per species for these analyses, choosing the sample with the least amount of missing data across the autosomes and Y chromosome, respectively. For the autosomal data, we sampled 10 loci of 5 kb each, requiring them to be located at least 10 kb from the nearest gene to minimize biases from selection, which were then treated as individual partitions in the analysis. We ran two independent MCMCTree runs, using the correlated rates clock model and sampling every 100 iteration after discarding the first 10,000 as burnin, until a total of 20,000 samples were retrieved. Due to the large heterogeneity expected across the autosomes from, e.g., rate variation or gene flow, we ran 10 replicates (i.e., sampling 10 new loci). The results of each run were first analyzed independently, and subsequently merged and

summarized across all runs. Effective sampling sizes (ESS) were obtained using tracer/1.7.1[66] (Supplementary Figs. 2, 3). For the Y chromosome divergence dating, using all sites resulted in poor convergence and ESS, which led us to use only SNPs that could be genotyped in all included samples (12,875 bp). We performed two independent runs, which converged to highly similar age estimates and reached ESS ≥ 323 (Supplementary Fig. 4).

### *TSPY* amplification, sequencing and analysis

Since whole genome sequencing data was available from a single male *C. denti* individual, we confirmed its Y-chromosome haplotype in four additional male *C. denti* by amplifying and sequencing a region of the Y-linked *TSPY* gene in two fragments, Y1 (497 bp) and Y2 (401 bp)[67] (Supplementary Fig. 5). Sanger sequencing was performed by the Molecular Cloning Laboratories (San Francisco, CA). Amplification was done with primer pairs 170 F:GGCGTCGTTGTGACCATTTG/691 R:GTG GTTTGGAATCTGACTGAGGTC (Y1) and 1710 F:AACTGTGGAGTCTT ATGCCCA/2160 R:GCATCTCCTCTGAACCACCAT (Y2), and sequencing with primer pairs 202 F:GAACGAGGGTGAGTTTCCACAG/667 R:A GAGCCTTGAGATGCAATGGGA (Y1) and 1745 F:TGTCCACACTAAC TGAGAAGTA/2119 R:ACTGCCTGCTGAGAAAAGACTACC (Y2).

Sequence chromatograms were inspected by eye and assembled using Geneious R11 11.0.5. We also used blast v.2.15.0 + [68] to identify the corresponding regions from the mapped whole genome sequences of the *C. denti* male and one of the two *C. wolfi* males. The two regions were then converted to fasta sequences and concatenated, complemented with publicly available guenon *TSPY* sequences[69,70] and aligned using MAFFT. After manual curation of the alignment, we constructed a median joining haplotype network using PopART/1.7[71].

### Nucleotide divergence and introgression statistics

To quantify excess autosomal allele sharing between *C. denti* and *C. mitis* (indicative of gene flow), we calculated D-statistics in Dsuite/0.4[72] using autosomal, biallelic SNPs. We used the rhesus macaque as outgroup, *mitis* group taxa as P3, *C. denti* as P2 and alternated between *C. mona*, *C. pogonias* and *C. wolfi* as P1. D-statistics was calculated for all possible combinations of samples, and we considered D-statistics significant if they differed from zero by more than three block-jackknife standard errors (Z-score > 3). We used Pixy/1.2.5[73] to calculate pairwise nucleotide divergence ($d_{XY}$) between pairs of species from the *mona*, *cephus* and *mitis* species groups in non-overlapping 50 Kb windows along the genome. Estimates of nucleotide divergence were scaled to coalescence time (t) assuming a generation time (g) of ten years and a mutation rate (μ) of 4.82e-9 substitutions per bp per, using the following equation:

$$t = (d_{XY}/(2*\mu))*g \qquad (1)$$

For *C. mitis*, we used only ssp. *opisthostictus* in these comparisons, since this lineage showed the closest Y-chromosomal relationship to *C. denti*.

To search for autosomal or X-chromosomal regions that might have introgressed from *C. m. opisthostictus* into *C. denti* alongside the Y chromosome, we constructed neighbor joining trees in sliding windows (non-overlapping 10 kb) along the genome, using PhyML/3.3[74]. Using a custom Python script, we identified windows that resembled the Y-tree topology in that *C. denti* was nested in a monophyletic *mitis* group clade as sister to *C. mitis*, to the exclusion of a monophyletic *mona* group clade (Supplementary Fig. 6). To investigate if genomic regions with a topology that resembled the Y-tree could have arisen through incomplete lineage sorting, we estimated $d_{XY}$ between *C. denti* and *C. wolfi*, and *C. denti* and *C. mitis* in the same, 10 Kb windows. We also quantified excess allele sharing between *C. mitis* and *C. denti* in these windows, by calculating the $f_d$ statistic using ABBABABA-windows.py (https://github.com/simonhmartin/genomics_general).

For this analysis, we used *M. mulatta* as outgroup, *C. mitis* as P3, *C. denti* as P2 and *C. wolfi* as P1.

### Identification of the pseudo-autosomal regions (PAR)

We searched for the pseudo-autosomal regions (PAR) on the X/Y chromosomes in the rhesus macaque reference genome using a coverage-based approach (Supplementary Fig. 7). Since the PAR are homologous and recombining, both females and males are expected to show a mapping coverage in this region that is similar to the autosomes, whereas males should only show half the autosomal mapping depth on the non-PAR X and Y chromosomes. We estimated the average mapping depth in non-overlapping 10 kb genomic windows using samtools/1.20[75], and normalized the values by dividing the depth of each window with the autosomal average.

### Inferences of demographic history with BPP-MSci

We used the multi-species-coalescent-with-introgression model (MSci) implemented in bpp/4.6.2[41] to infer divergence times, ancestral population sizes and proportion of gene flow among focal species. In this analysis, we included *C. denti*, *C. wolfi*, *C. pogonias*, *C. mona*, *C. neglectus*, *C. cephus*, *C nictitans*, *C. mitis opisthostictus* and *M. mulatta*, choosing the sample with the least amount of missing data on the Y chromosome in taxa with more than one available sample. We sampled a total of 1000 loci, each 1000 bp long, requiring them to be at least 10 kb from the nearest gene and 50 kb apart to avoid effects of selection and linkage, respectively. We used the topology from the ASTRAL analyses, and added three unidirectional migration bands: (1) from the ancestor of the *mitis* lineage into the ancestor of *C. pogonias*, *C. wolfi* and *C. denti*, (2) from the ancestor of the *cephus* lineage into the ancestor of *C. pogonias*, *C. wolfi* and *C. denti*, and (3) from *C. mitis* into *C. denti*. The program was set to run 20,000 iterations as burnin, and then sample every second MCMC iteration until 200,000 samples were collected. Two independent runs were performed with the same parameters, and after confirming that they converged on similar estimates, they were merged and analyzed jointly. The output tree was scaled to years and the theta values converted to effective population size (Ne), using the bppr package[76] assuming a mutation rate of 4.82e-9 substitutions per bp per generation, and a generation time of 10 years[31]. BPP assigns an Ne estimate to all branches, including those leading to hybrid nodes (e.g., the terminal branch of *C. denti* had two Ne estimates, one before and one after the incoming gene flow from *C. mitis*). To simplify our demographic model for visualization and downstream simulations, we calculated a single Ne estimate using the harmonic mean of Ne values along all branches affected by gene flow.

### Coalescent simulations with msprime

**Autosomal simulations.** To explore the probability of drift leading to the fixation of the introgressing Y chromosome, we ran coalescence simulations using msprime/1.2[42]. We simulated the demographic history inferred with BPP-MSci (Fig. 5), with varying amounts of gene flow from *C. mitis* into *C. denti*. For the autosomal simulations, we used the effective population size (Ne) estimates directly from the BPP output and simulated 100 loci of 1 Mb each, always including a single pulse of 2.1% migration from the *mitis* group ancestor into the ancestor of *C. denti*, *C. pogonias* and *C. wolfi* at 310,000 generations ago (event 1 in Fig. 5), and a pulse of 8.5% migration from the *cephus* ancestor into the same recipient population at 270,000 generations ago (event 2 in Fig. 5). The third migration pulse, from *C. mitis* into *C. denti* (event 3 in Fig. 5), was set to occur 100,000 generations ago (10,000 generations after the *C. denti* split from *C. wolfi*), with varying proportions from 0 to 1 %. We then used Dsuite to calculate D-statistics (P1 = *C. wolfi*, P2 = *C. denti*, P3 = *C. mitis*, Outgroup = *M. mulatta*) and associated Z-scores. The D-statistics from simulated data were used to identify the upper limit of plausible migration proportions from *C. mitis* into *C. denti*

(assuming that gene flow was a neutral process), given our empirical estimates.

**Y chromosome simulations.** We simulated a non-recombining 10 Kb locus to mimic the Y chromosome under the same demographic history as above, dividing the effective population sizes by four since the ratio $Ne_{Ychrom} / Ne_{Autosomes} = 0.25$[77] (Supplementary Fig. 11). Considering that guenon males disperse more than females, it is likely that gene flow is predominantly driven through male migration. Under the conservative assumption that only males migrate, the effective Y-chromosomal migration rate would be twice that of the autosomes. Therefore, we doubled the migration proportions from *C. mitis* into *C. denti* compared to the autosomal simulations, to make them directly comparable. We simulated 100 replicates of 1000 simulations per migration rate (0–2% with a stepwise increase of 0.1%), and counted the number of times *C. denti* and *C. mitis* formed a monophyletic clade.

**Pseudo-autosomal region (PAR) simulations.** We also used msprime simulations to explore the effect of Y chromosome introgression on pseudo-autosomal region (PAR, Supplementary Fig. 8D). To this end, we simulated genomes consisting of one 50 kb non-recombining locus (resembling the Y chromosome), a PAR-like region of 150 kb, and three autosomal loci of 50 kb each. We used the same demography as for the Y chromosome introgression simulations described above and used an autosomal recombination rate of 4.48e-9 per bp and generation[78]. Along the PAR, we used a recombination rate of 1, 10 or 20 times the autosomal rate, and simulated 10,000 replicates per PAR recombination rate with 1% introgression from *C. mitis* into *C. denti*. We discarded simulations where the introgressing Y chromosome was lost, and calculated $f_d$ (P1 = *C. wolfi*, P2 = *C. denti*, P3 = *C. mitis*, Outgroup=*M. mulatta*) in 50 kb windows in remaining simulations.

**Forward simulations with SLiM**
To test how selection affects the probability of Y chromosome fixation in *C. denti*, we ran forward simulations in SLiM[45]. We simulated a single population of 200,000 individuals for 100,000 generations (mimicking the *C. denti* lineage after the split from *C. wolfi*) and used the built-in functionality of modeling a Y chromosome to track the frequency of a novel Y-linked allele introduced in the first generation. We tested selection coefficients (s) ranging from 0–0.01, and initial Y chromosome frequencies (equivalent to incoming male migration) in the range of 0.1–1% (range informed by previous neutral simulations, Fig. 6A). We ran 100 replicate simulations for each frequency and selection coefficient, counting the number of times the allele was lost, fixed or still segregating after 100,000 generations.

**Identifying candidate genes under selection and possible drive elements**
We used several approaches to explore putative genes under positive selection that might have driven the introgressing Y chromosome to fixation in *C. denti*. First, we identified protein coding genes annotated on the rhesus macaque Y chromosome which showed differentially fixed amino acid sequence in *C. mitis opisthostictus* and *C. denti* compared to *C. wolfi* and *C. pogonias* (Supplementary Data 4). We used custom Python scripts to translate the transcripts from each gene into amino acid sequences and to count the number of fixed differences for each gene, after excluding genes with internal stop codons in focal lineages. Second, we tested whether a model of adaptive or neutral evolution along the *C. denti* and *C. mitis* branch was a better fit for genes with fixed amino acid differences, using codeml in paml/4.9j and HyPhy/2.5.51[79] (Supplementary Data 5). Codeml was run using the branch-site model: A model allowing for positive selection on a subset of sites along a specified set of foreground branches was compared to a model of neutral evolution. We tested the following foreground branches, based on

the Y-chromosomal topology: (1) the ancestral *mitis* group branch and all descendants (including *C. denti*); (2) the ancestral *mitis* group branch, the ancestral *C. mitis* + *C. denti* branch, and the terminal branches of *C. mitis* and *C. denti*; (3) the ancestral *C. mitis* + *C. denti* branch and their respective terminal branches; and (4) only the terminal branch of *C. denti*. For these analyses, we used a single individual for each *Cercopithecus* species in our data set and *Chlorocebus sabaeus* as the outgroup, choosing the sample with the least amount of missing data. HyPhy was run using the 'meme' algorithm (Murell et al. 2012), on a concatenated alignment of the Y chromosome genes using the same species.

We also identified sex-chromosomal regions with increased coverage, indicative of copy-number expansions, as putative candidates for meiotic drive elements[22,47] (Supplementary Figs. 12–15). To this end, we calculated the average mapping depth in sliding windows of 5 Kb (step size 1 Kb) along the rhesus macaque X and Y chromosomes. The coverage was normalized for each sample and chromosome by dividing the window depths by the chromosome-wide average. Next, we calculated the coverage ratio of *C. denti* to *C. wolfi*, alternating through all pairwise comparisons of individuals between these species. If meiotic drive through a copy number expansion of specific Y-linked genes occurred in *C. denti*, such regions are expected to show higher coverage in *C. denti* genomes relative to its sister *C. wolfi*. In this case, we also expect to observe compensatory increase in copy number on the X-chromosome.

### Reporting summary
Further information on research design is available in the Nature Portfolio Reporting Summary linked to this article.

## Data availability
Whole genome sequencing data generated for this project are available at ENA under accession number PRJEB73870, with individual accession codes listed in Supplementary Data 1. Previously published whole-genome sequencing data used in this project are accessible at SRA under project accession numbers PRJEB67744 [https://www.ncbi.nlm.nih.gov/bioproject/1031930], PRJEB32105 [https://www.ncbi.nlm.nih.gov/bioproject/572824], PRJNA240242, PRJNA251548, PRJNA595456, and PRJNA512907, with sample specific accession codes listed in Supplementary Data 1. The *TSPY* sequence data generated for Supplementary Fig. 5 are available as a Source Data file, and at GenBank under accession codes PQ570681, PQ570682, PQ570683, PQ570684, PQ570685, PQ570686, PQ570687, PQ570688 (Supplementary Data 2). Previously published *TSPY* used in Supplementary Fig. 5 is available through accession codes EF517803.1, AY450876.1, AY048057.1, AY450874.1, EF517804.1, AY897616.1, AY665648.1, AY450878.1, AY048058.1, AY450877.1, EF517805.1, AY048059.1, AF284281.2, AY450880.1, AY450875.1, JN106053.1, JN106052.1, and EF517806.1. The *M. mulatta* reference genome is available under the NCBI RefSeq accession number GCF_003339765.1. Source data are provided as a Source Data file. Source data are provided with this paper.

## Code availability
Analytical scripts used in this project are available at https://github.com/axeljen/denti_ychrom_scripts[80].

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

## Acknowledgements

We thank Stuart JE Baird for useful discussions on mammalian Y chromosome introgression, Terese Hart for logistical support during sample collection in the Democratic Republic of Congo, and Louis Rugyerinyange, Felix Mulindahabi and Beth Kaplin for logistical support for sample collection in Nyungwe National Park, Rwanda. We acknowledge Faustin Kahindo, Maurice Emetshu, Gilbert Paluku, Peter-Philip Niehoff and James Gray for assistance with sample collection in the field. We thank the Congolese Institute for the Conservation of Nature (ICCN) and the Rwanda Development Board (formerly ORTPN) for research permissions. Sequencing was performed by the SNP&SEQ Technology Platform in Uppsala. The facility is part of the National Genomics Infrastructure (NGI) Sweden and Science for Life Laboratory. The SNP&SEQ Platform is also supported by the Swedish Research Council and the Knut and Alice Wallenberg Foundation. The computations were enabled by resources in projects SNIC 2022/6-325 and SNIC 2022/5-561, provided by the Swedish National Infrastructure for Computing (SNIC) at Uppsala University (UPPMAX), partially funded by the Swedish Research Council through grant agreement no. 2018-05973. The project was supported by the Swedish Research Council VR (2020-03398) grant to K.G., Zoologiska Stiftelse grants to A.J., Margot Marsh Biodiversity Foundation and FAU Foundation, Inc. to K.M.D. A.T. acknowledges funding from NSF award #1718715.

## Author contributions

K.G., K.M.D., and A.J. designed the study. K.M.D., J.A., and J.A.H. led sample collection in the field. K.M.D., E.R.H. and A.J.T. performed DNA extractions, prepared and analyzed Sanger sequencing data. A.J. processed and analyzed whole genome data with assistance from S.P. K.G. and K.M.D. acquired funding. A.J. and K.G. wrote the manuscript with contributions from K.M.D. All authors reviewed and approved the manuscript.

## Funding

## Competing interests

The authors declare no competing interests.

## Additional information

**Supplementary information** The online version contains Supplementary Material available at https://doi.org/10.1038/s41467-024-54719-8.

