## [Peer Review file · Nature Communications]

Y chromosome introgression between deeply divergent primate species

Corresponding Author: Mr Axel Jensen

Version 0:

Reviewer comments:

Reviewer #1

(Remarks to the Author)

I really enjoyed reading the manuscript "Breaking the rule: An exceptional Y chromosome introgression between deeply divergent primate species". The manuscript presents strong evidence of a Y-chromosome historical introgression event among distantly related guenon species. It also investigates the extent of selection required for the fixation of the introgressed Y chromosome to have occurred. The data are adequate, and the analyses are comprehensive. I believe this study will be of great relevance for audiences broadly interested in understanding the processes and mechanisms of hybridization and its evolutionary consequences in natural populations. The manuscript is also well-written. I found some minor typographic errors and made a few editorial suggestions that I outlined below by line number. Besides those minor changes, I also have a few comments that I sincerely hope authors will find useful:

1. Throughout the manuscript there is a strong statement about the results being an exception to Haldane's Rule. I agree that overall, the introgression of a Y-chromosome may be at odds with the expected pattern of Haldane's rule. However, the authors quote Haldane's Rule as "when in the offspring of two different animal races one sex is absent, rare, or sterile, that sex is the heterozygous sex". In my view, this quotation focuses on the outcome at the moment of hybridization, but since this manuscript focuses on a historical introgression event, I think other explanations could also be feasible. For example, in natural hybrid zones interbreeding does not only occur between non-admixed individuals of different species, but also between admixed and non-admixed individuals (backcrossing), and between multigenerational hybrids. It could be that Haldane's Rule remained true when non-admixed individuals of the two taxa first interbred, producing only fertile females in the first generation(s) of hybridization, but after backcrossing and multigenerational hybrid interbreeding, males might have been produced if the right alleles met again in a single individual and restore their "compatibility". If these gene combinations provided a strong selective advantage to those admixed individuals, and they preferentially backcrossed with individuals in the *C. denti* lineage, then it might be possible that the Y chromosome could have been fixed in that population. Multiple generations of backcrossing (through ~ 2 million years of evolution) could minimize the signature of neutral autosomal introgression. This scenario would still be consistent with the observed (and very interesting) results presented in the manuscript, even if Haldane's Rule was present when the two species first hybridized. I know this scenario is less parsimonious, but based on observations in current natural hybrid zones, I think it might be plausible. I suggest authors to be more explicit as to why they consider their findings to be "an exception" to Haldane's Rule.

Also, In the introduction authors only talk about the "dominance theory" of Haldane's Rule, but in the manuscript, they also explore other potential causes, such as meiotic drive. There are several other potential explanations that have been postulated for Haldane's Rule, but authors did not expand on any other. Because of the strong emphasis in the on Haldane's Rule, I expected to read more discussion on the alternative hypotheses (e.g., faster-male evolution, faster X, Y incompatibilities, etc.). At the least, I think authors should bring meiotic drive to the introduction. I would also recommend including a brief mention of alternative mechanisms to Haldane's Rule and to explain why they decided to focus on those explored in the study. It would also be worth to acknowledge in the discussion the potential role of other mechanisms.

2. I found the analyses of introgression of the X chromosome very interesting, but I expected some discussion about the regions in the X that are inferred to have been introgressed from *mitis*. For example, are the sequences of these regions more similar between *C. denti* and *C. m. opisthostictus* when compared to other *mitis* group individuals? Do we know if any of these introgressed regions are part of the PAR region of the X chromosome? Are there regulatory elements/genes in

these regions known to interact with the Y?

3. Figure 1. The autosomal (ASTRAL) tree does not have branch support values to fully evaluate the support of the topology. Given that this is a critical component of the manuscript I suggest including them.

4. Figure 2. Please provide the reference(s) for the species distributions used in the figure.

5. Figure 6B. Please explain in the legend what the gray vs. black symbols denote. Also, it would be easier to see if instead of black/gray you used filled/non-filled circles as in A, because it is hard to differentiate between overlapping gray circles from black circles.

6. Authors provide a statement about following proper country-specific research and export permits (and their numbers in Table S1, which is great!). Given that all primates are protected by CITES I will encourage authors to also include a statement about exportation/importation from the US to Sweden if needed.

Other minor editorial suggestions:

Line 46. Add a brief definition of “genomic incompatibilities”. Are these BDM? Perhaps add a reference?

Line 62. Change “these incompatibilities” for “the incompatibilities between loci in the Y-chromosome and autosomes”.

Line 74. This part could be more explicit, as it is unclear what “All this” mean (i.e., how the fact that many species hybridize, and that males disperse and females are philopatric “makes guenons an ideal system to study Haldane’s Rule along a speciation continuum”). Maybe there is something missing?

Line 76. Add reference to the statement.

Line 149 and beyond. In the text the pairwise nucleotide divergence is abbreviated as d_{xy} , but in some figures (Figs. 3C and S6) it is written in capital letters with XY as subscript. Please check for consistency throughout the manuscript.

Line 151. I suggest adding “... Y chromosome must have been polymorphic in the ancestral population that gave rise to the mona and mitis/cephus lineages and remained polymorphic and unsorted along the branches of these groups.”

Line 181. Add “autosomal” between “excess” and “allele”.

Line 186. Throughout the manuscript you typically use “ref#” to cite work previously published, but here you used “Jensen et al. (2023)”. Please check for consistency.

Line 189. Add “likely” between “gene flow event” and “occurred”, as that is an inference based on previous analysis.

Line 197. I suggest adding “...in comparison to allele sharing between the later and *C. pogonias*...” to make the comparison clearer.

Line 211 and beyond. Like the abbreviation of d_{xy} , there is inconsistency in the abbreviation of F_D , as it is sometimes abbreviated as f_D (e.g., line 581). Please check for consistency throughout the manuscript.

Line 213. It says, “...55 protein coding genes (Table S4)...”, but the table only shows 54.

Line 235. I suggest using “support”, instead of “confirm”, as these are inferences.

Line 335. It would be useful for the reader if it is more explicit what “genotyped sites” means in this sentence.

Line 423. Provide the example and the reference of this “notable exception”.

Line 430. Change “showed” to “would have had”, as you are only inferring this divergence.

Line 435. I suggest using “strongly”, instead of “unambiguously”, as these are estimates based on simulations.

Line 458. I suggest mentioning that in the orangutan study the expression levels were similar, and later in the paragraph discussing the type of expression data that would be needed to address this possibility in guenons.

Line 483. Consider here the possibility that males were not produced (or were infertile) in the first generations of hybridization, as described in my initial comment.

Line 491. Include between brackets the number of samples used in this study for each taxon.

Line 537. Please be more explicit as to what “converting the genotypes to a chromosome-wide alignment” means. Does it mean that you first aligned your sequences to the reference *Macaca* Y chromosome?

Line 559. The four individuals for which the TSPY gene was Sanger sequenced are not included in Table S1. Please either add them to the table and/or provide information of their precedence (wild, captive, location, permits, etc.).

Line 563. Please explain how you “extracted the corresponding regions” and, if relevant, provide the code.

Line 593. Are these “consecutive windows” only those that showed signatures of introgression?

Line 677. For clarity, add “...with the least amount of Y chromosome missing data”.

Reviewer #2

(Remarks to the Author)

(also sent as attachment, since the equation may not read easily here)

In this manuscript, Jensen et al. detect a Y chromosome introgression event between two deeply diverged guenon species, namely from *C. mitis* into *C. denti*. The authors first reconstruct the phylogenetic tree for 22 previously reported guenon genomes, plus two newly sequenced species, detecting a discrepancy between the autosomal and the Y chromosomal phylogenetic trees. This discrepancy is unlikely to be caused by incomplete lineage sorting (ILS), since the Y chromosome shows much lower divergence than autosomes. D-statistics suggest low levels of autosomal introgression from *mitis* into *denti*, implying that the Y chromosome introgressed at low frequency. The authors then perform extensive simulations with and without selection, concluding that drift alone is unlikely to have driven the introgressed Y chromosome to fixation. The authors pose two possible explanations for selection acting on the Y chromosome, namely adaptive introgression and meiotic drive.

The rationale and results of the paper are well explained and well-organized, and finding of Y chromosome introgression between unusually deeply diverged species is an interesting case study for the speciation/admixture genetics community (as exceptions to the rule often help us understand the rule itself!). We agree that the scenario favored by the authors is currently the most reasonable explanation. However, we have several suggestions that we hope will help improve the manuscript overall.

Major comments

1) Much of the argument for selection comes from simulations in msprime and SLiM. However, we note that they can be entirely replicated using analytical methods alone, with the advantage of removing stochasticity, providing direct estimations of very low probabilities, and linking the results directly to population genetic theory. While we understand that researchers differ in their attraction to simulation-based logic versus analytical logic, consider adding and/or replacing the simulations for the ILS analysis and the selection analysis (Figure 6C) following the suggestions below. For 6B (see below), we think simulations are not necessary at all since the probability of fixation of a neutral mutation is very well-worked out and familiar to most population/evolutionary geneticists.

First, using standard coalescent theory, one can calculate the probability that the Y chromosomes of *wolfi* and *denti* do not coalesce before reaching the split time with *mitis* clade. Taking the demographic parameters from the model inferred using BPP-MSci (fig. 5):

$$\Pr(\text{no coalescence between } wolfi \text{ and } denti) = \exp(-t/(2N_e/4)) = \exp\left[\frac{-600,000}{(2 \times 200,000/4)}\right] = 0.0025$$

Where $2N_e/4$ is the effective population size of the Y chromosome, t is the time in generations between the *wolfi/denti* split and the *denti/mitis* split. Furthermore, in order for *denti* and *mitis* to coalesce deep in time due to ILS, this probability is further reduced by 1/3, meaning that there is a less than 0.1% chance that the Y chromosomal discrepancy is due to ILS. This is probably why the authors do not observe any of their 1000 simulations grouping *denti* and *mitis* together in Fig. 6B when Y chromosomal migration is 0 (but the probability is not 0!).

Having established that ILS is unlikely, we can assume (in the absence of selection) that the initial Y chromosomal frequency equals the probability of fixation. This explains the one-to-one relationship between the migration rate and the probability of *denti/mitis* monophyly in Fig. 6B. The uncertainty around the median is simply due to the stochasticity of sampling from a binomial of size 1000, with probability of success equal to the initial frequency. More simulations would decrease the uncertainty around the mean, but the expectation remains the same. Thus, Fig. 6B is not necessary, since it's based on the well-known finding that the probability of fixation under drift is equivalent to its original frequency.

Finally, Fig. 6C can also be replicated without the need for performing simulations. The procedure is a bit more complex, and involves using binomial sampling with selection for a Wright-Fisher population together with phase-type theory to calculate the probability of fixation based on the initial frequency and the selection coefficient. The procedure is explained in section 4.4 and Fig. 5 in Hobolth et al. 2024 (<https://doi.org/10.1016/j.tpb.2024.03.001>). We attach accompanying R code to replicate Fig. 6C of the manuscript using this approach. Regardless of what you chose to do, if simulations remain a main source of inference, please provide the code for replicating the SLiM and/or msprime runs.

2) The authors have not explicitly characterized the pseudoautosomal region (PAR). This is the only recombining region between the X and the Y chromosomes, and it might show interesting patterns of divergence or mutation (see Bergman and Schierup 2022, <https://doi.org/10.1186/s13059-022-02784-x>). It is also unclear whether PAR has been included in the

divergence calculations, which might bias some of the inferences. A Y-chromosome-wide plot showing the areas included in the analyses would be helpful to clarify this point; there is a coverage plot in the supplement, but it's a bit hard to interpret, since it's based on coverage normalized to the mean.

3) The manuscript comes down strongly in favor of a selective hypothesis (e.g., "Our analyses unambiguously suggest..."). We agree that this is a likely possibility, but the language of certainty should be modified throughout (and the authors should clarify that they consider the meiotic drive possibility a subset of selection here; or perhaps refer to the combination of drive and adaptive introgression as 'non-neutral processes'). Part of the uncertainty involves use of point estimates for divergence dates and allele frequencies. For example, the introgressed Y is treated as fixed, but this is based on only 5 males. Apparent fixation with a sample of 5 is also compatible with high, but not fixed, frequencies, which would change the simulation estimates. Similarly, the split times (8 million years ago for autosomal divergence, 1.9 mya for Y) are treated as fixed, but both likely have uncertainty estimates around them that affect how "necessary" selection is to the story. And of course, clearly corroborative functional evidence has not yet been obtained.

Minor comments

In Fig. 3C, the title of the plot states "Nucleotide diversity on the Y chromosome", but consider editing the title, since you also show the diversity on the autosomes alongside the Y chromosome.

Can the distribution of sizes of the putatively introgressed regions on the autosomes be used to estimate gene flow dates here as well? It would be helpful to see if the autosomal dating corroborates the timing estimated for the Y.

Reviewer #3

(Remarks to the Author)

The authors study 57 samples of guenon species. These sequenced reads were mapped against the rhesus macaque reference genome.

They find a discrepancy in the Y chromosome tree vs the species tree between *C. Denti* and *C. Mitis* subspecies. The divergence of the Y chromosomes is >6 millions years. This is very exciting find and to my knowledge the most divergent Y chromosome introgression that we know of. It provides a notable exception to Haldanes rule. The autosomal divergence is 8 million years and the Y chromosome divergence is 2 million years. The authors confirm this finding using of TSPY genes in four additional *C. denti* males.

The authors conduct simulations to show that this is highly unlikely under neutrality and conclude that there must be some selection going on.

The look for a plausible mechanism for this and detect non-synonymous differences on the Y chromosome in 2 genes (KDM5D and USP9Y) and find a large region (1.35 Mb) has higher copy number suggestive of gene duplications - from humans we know these are very common but we do not know the function of them yet.

Overall I enjoyed the paper and find the conclusions well supported. Good job!

I only have a few comments:

1. In the section beginning at line 144 the authors want to distinguish between ILS and introgression using the divergence between the autosomes and Y chromosomes. They use the divergence measure D_{xy} - could you please provide units for this (mutations per basepair I assume)? In addition, I think it could be really useful to convert the D_{xy} on the y axis values in Figure 3 to coalescence times in years to make it easier to compare to Figure 1

2. Dstatistics

The informative statistic here to test for gene flow between *C. mitis* and *C. denti* would be on the form: (*C. Wolfi*, *C. Denti* ; *C. mitis*, rhesus macaque)

There is only support for geneflow in some individuals and I found table S3, D-statistics very helpful. However I am confused about the columns BBAA, and f4-ratio. Why do we need this information? And if there was an F4 ratio test what f4-values are being divided? I couldn't find this in the supplement but perhaps I missed it.

3. The F_d statistic and in combinations with d_{xy} and 4 different filters seems very ad-hoc to me. How accurate is this method - how does it work with the data you have? If the authors want to use this to detect introgressed regions they should show how well it works for instance the accuracy and false positive rate of this statistic using simulated data under their inferred demography. Alternatively this section can be taken out of the paper.

Minor comments

Figure 1 - Can the authors please provide bootstrap values on the tree or some measure of variance. They write in the text that the grouping of *C. mitis* and *C. albigularis* is poorly resolved so that should be indicated in Figure 1.

Fig 1 vs Fig S1

You use salmon in Fig S1 instead of red - but the colors look the same on my computer. Maybe just a typo

line 234 and line 615 - needs units for the mutation rate

line 432 what is the unit of divergence? 10.8% divergence between human and chimpanzee (ref 39) is for mitochondrial! Nuclear divergence is more like 1-2% I believe.

Reviewer #4

(Remarks to the Author)

Version 1:

Reviewer comments:

Reviewer #1

(Remarks to the Author)

The authors have carefully addressed all my comments to the original manuscript. I have no further comments.

(Remarks on code availability)

Reviewer #2

(Remarks to the Author)

This is the second time we have seen this manuscript. As before, we continue to find the results interesting, and the scenario favored by the authors (introgression of a deeply diverged Y chromosome, followed by selection or meiotic drive to drive it to high frequency/fixation) plausible.

The revisions in this version improve the manuscript overall, although most of the changes are only clarificatory in nature. Engaging with the additional analyses or alternative explanations suggested by the reviewers could have strengthened the manuscript even further (e.g., alternative mechanisms that give rise to Haldane's Rule; the possibility that the results are not an exception to Haldane's Rule as typically interpreted—which the authors agree with in the rebuttal, but still appears in the title; analytical—as opposed to only simulation-based—support for the selection scenario).

One key addition is the attention to the PAR region of the X. Here, the authors now report that the PAR shows greater divergence between *C. mitis* and *C. denti* than on the rest of the X or the autosomes. While this result is consistent with patterns reported in the great apes in the absence of Y chromosome introgression, it is somewhat puzzling given that this region specifically is expected to recombine with the putatively introgressed *mitis* Y. In the scenario proposed by the authors, shouldn't the region therefore exhibit reduced divergence between *C. mitis* and *C. denti* (in the absence of selection against the introgressed *mitis* DNA)? To put this result in context, the authors could also have calculated the divergence between *C. denti* and *C. wolfi* on the X PAR and compared it with the divergence between *C. denti* and *C. mitis* (as shown in Figure S8). The *denti/wolfi* comparison would serve as the "baseline" case (i.e., no introgression but the same evolutionary history of divergence as *denti/mitis*).

Minor comment:

- The response to reviewers indicated that Figure 6B, which shows the relationship between initial allele frequency and fixation under neutrality, had been moved to the supplement as Figure S11, but it remains in the main text of the revised manuscript that we saw. As noted in the initial review, this is a trivial result since this relationship is long-established by theory. In addition, the exact relationship between autosomal migration and initial Y chromosome frequency assumes that all migrants are males, which is not particularly well-justified.
- It would be useful for the reader to refer to relevant supplementary figures in the methods section (such as referring to Figure 7B in the paragraph starting in line 614).
- In line 313, consider providing the probability for Y chromosomal introgression + ILS using analytical derivations.

(Remarks on code availability)

The instructions on how to run the code are clear and straightforward. The code can be run smoothly, and results are reproducible.

Reviewer #3

(Remarks to the Author)

The authors have addressed all my comments. The manuscript is in great shape and I will be happy to see it published! Well

done to the authors!

- My last comment (and this is just a suggestion) would be to change Figure 4 slightly. My issue is that the authors are performing a f4 test (or D-statistic) with rhesus macaque as outgroup. However in the Figure they only show the 3 populations that change and it appears they are doing a f3 test at first glance. I would recommend adding the P4 population to the facets in the figure or at least highlight it in the figure text.

(Remarks on code availability)

Very well documented github repository!

Reviewer #4

(Remarks to the Author)

(Remarks on code availability)

Version 2:

Reviewer comments:

Reviewer #2

(Remarks to the Author)

The authors have satisfactorily addressed all of our remaining concerns (and thank you for sharing the thorough and convincing PAR analysis).

(Remarks on code availability)

We did not look at the code again after the previous revisions, as it did not appear to change.

Reviewer #4

(Remarks to the Author)

(Remarks on code availability)

REVIEWER COMMENTS

Reviewer #1 (Remarks to the Author):

I really enjoyed reading the manuscript “Breaking the rule: An exceptional Y chromosome introgression between deeply divergent primate species”. The manuscript presents strong evidence of a Y-chromosome historical introgression event among distantly related guenon species. It also investigates the extent of selection required for the fixation of the introgressed Y chromosome to have occurred. The data are adequate, and the analyses are comprehensive. I believe this study will be of great relevance for audiences broadly interested in understanding the processes and mechanisms of hybridization and its evolutionary consequences in natural populations. The manuscript is also well-written. I found some minor typographic errors and made a few editorial suggestions that I outlined below by line number. Besides those minor changes, I also have a few comments that I sincerely hope authors will find useful:

We thank the reviewer for the overall positive assessment of our work.

1. Throughout the manuscript there is a strong statement about the results being an exception to Haldane’s Rule. I agree that overall, the introgression of a Y-chromosome may be at odds with the expected pattern of Haldane’s rule. However, the authors quote Haldane’s Rule as “when in the offspring of two different animal races one sex is absent, rare, or sterile, that sex is the heterozygous sex”. In my view, this quotation focuses on the outcome at the moment of hybridization, but since this manuscript focuses on a historical introgression event, I think other explanations could also be feasible. For example, in natural hybrid zones interbreeding does not only occur between non-admixed individuals of different species, but also between admixed and non-admixed individuals (backcrossing), and between multigenerational hybrids. It could be that Haldane’s Rule remained true when non-admixed individuals of the two taxa first interbred, producing only fertile females in the first generation(s) of hybridization, but after backcrossing and multigenerational hybrid interbreeding, males might have been produced if the right alleles met again in a single individual and restore their “compatibility”. If these gene combinations provided a strong selective advantage to those admixed individuals, and they preferentially backcrossed with individuals in the *C. denti* lineage, then it might be possible that the Y chromosome could have been fixed in that population. Multiple generations of backcrossing (through ~ 2 million years of evolution) could minimize the signature of neutral autosomal introgression. This scenario would still be consistent with the observed (and very interesting) results presented in the manuscript, even if Haldane’s Rule was present when the two species first hybridized. I know this scenario is less parsimonious, but based on observations in current natural hybrid zones, I think it might be plausible. I suggest authors to be more explicit as to why they consider their findings to be “an exception” to Haldane’s Rule.

We thank the reviewer for this input, and agree that this is an important point to include as a possibility. We included this in the discussion (ll. 450-452), and carefully reworded throughout the manuscript to reflect that a fertile F1 hybrid male is not a necessity for the Y chromosome introgression (e.g., abstract, ln. 507). However, even in the event that introgression occurred via an admixed population and multigenerational hybrids, for the Y chromosome to penetrate the species boundary, a fertile crossing of a male that is predominantly *mitis* and hence carries the *mitis* Y-chromosome and and a female that is predominantly *denti* is still required,. The reviewer also rightly points out that Haldane’s rule classically applies to the outcome at the moment of hybridisation. However, the outcome of Haldane’s rule acting early in the hybridisation process is the congruence between autosomal and Y-chromosomal trees, as we highlight in the introduction on ll. 67-71. In our system, we observe a pattern that contradicts this expectation.

Also, In the introduction authors only talk about the “dominance theory” of Haldane’s Rule, but in the manuscript, they also explore other potential causes, such as meiotic drive. There are several other potential explanations that have been postulated for Haldane’s Rule, but authors did not expand on any other. Because of the strong emphasis in the on Haldane’s Rule, I expected to read more discussion on the alternative hypotheses (e.g.,

faster-male evolution, faster X, Y incompatibilities, etc.). At the least, I think authors should bring meiotic drive to the introduction. I would also recommend including a brief mention of alternative mechanisms to Haldane's Rule and to explain why they decided to focus on those explored in the study. It would also be worth to acknowledge in the discussion the potential role of other mechanisms.

As the reviewer rightly points out, there are several other commonly invoked theories for Haldane's rule besides the dominance theory. We have expanded our examples in the introduction to provide more context to these theories (II. 53-63). We now also included meiotic drive in the introduction (II. 74-75). However, we note that the manuscript is not discussing meiotic drive as a cause of Haldane's rule (although it has been reported to act as such too), but rather as a potential mechanism to overcome the negative selection on heterogametic hybrids/admixed individuals expected under Haldane's rule. Overall, we think that discussing the mechanisms of Haldane's rule is beyond the scope of this manuscript as we cannot test for them in our system. We therefore prefer to retain a more general background on Haldane's rule in the introduction, without going into the specifics, which are not further discussed in the study.

2. I found the analyses of introgression of the X chromosome very interesting, but I expected some discussion about the regions in the X that are inferred to have been introgressed from *mitis*. For example, are the sequences of these regions more similar between *C. denti* and *C. m. opisthostictus* when compared to other *mitis* group individuals? Do we know if any of these introgressed regions are part of the PAR region of the X chromosome? Are there regulatory elements/genes in these regions known to interact with the Y?

Following the suggestion of reviewer 3 (see below), we have replaced the fd/dxy outlier based analyses of autosomal/X-chromosomal introgression, with a topology based approach (II. 229-238). This method is arguably less arbitrary and less prone to false positives, but also identifies considerably fewer genomic segments as introgressed (only 110 kb across the genome, in line with the almost non-detectable signal from the D-statistic). We identified four genes that overlapped these putatively introgressed windows (none of which is located on the X chromosome). We agree that it would be very interesting to know more about introgressed segments on the autosomes and the X-chromosome, but given the age of the event and levels of the introgression these segments are likely too short and too few to confidently identify (other than the ones we identified with the topology based approach). We have also included an additional analysis identifying the PAR region, which is only assembled on the reference macaque X chromosome and conduct additional analyses for this region (II. 240-248, Figures S7-S8). To directly answer the reviewer's question: Our original approach identified three X-chromosomal regions (each 10 kb) as putatively introgressed. None of them were located in the PAR region, and they did not overlap any annotated genes in the macaque reference genome. With the topology-based method, no introgressed regions are detected on the X-chromosome.

3. Figure 1. The autosomal (ASTRAL) tree does not have branch support values to fully evaluate the support of the topology. Given that this is a critical component of the manuscript I suggest including them.

We added the local posterior branch support to Figure 1. Please note that all genera, species groups, and species receive full support (II. 105-106).

4. Figure 2. Please provide the reference(s) for the species distributions used in the figure.

We downloaded the distribution ranges from IUCN, which is now appropriately cited in the figure legend (ln. 151).

5. Figure 6B. Please explain in the legend what the gray vs. black symbols denote. Also, it would be easier to see if instead of black/gray you used filled/non-filled circles as in A, because it is hard to differentiate between overlapping gray circles from black circles.

Following a suggestion from reviewer 2, this panel has been replaced in the main figure. The original Figure 6B has been moved to the supplementaries (Figure S11). The confusion that the reviewer raises here is due to the fact that we're using semi-transparent circles in this figure, which when stacked on top of each other may look like different colors (black/gray), but all circles have the same color.

6. Authors provide a statement about following proper country-specific research and export permits (and their numbers in Table S1, which is great!). Given that all primates are protected by CITES I will encourage authors to also include a statement about exportation/importation from the US to Sweden if needed.

Tissue samples were exported following CITES and country specific regulations, as stated in the ms and table S1, and DNA was extracted in the country of import. DNA extracts were then transferred to SciLifeLab in Sweden which provides sequencing services for research purposes. This, to the best of our knowledge, does not need any specific clearance.

Other minor editorial suggestions:

Line 46. Add a brief definition of "genomic incompatibilities". Are these BDM? Perhaps add a reference?

We restructured this part of the introduction, following a previous suggestion from the reviewer and have added the reference to BDMI (ll. 53-63).

Line 62. Change "these incompatibilities" for "the incompatibilities between loci in the Y-chromosome and autosomes".

After the restructuring of these paragraph (see previous reviewer point) this sentence has been removed.

Line 74. This part could be more explicit, as it is unclear what "All this" mean (i.e., how the fact that many species hybridize, and that males disperse and females are philopatric "makes guenons an ideal system to study Haldane's Rule along a speciation continuum"). Maybe there is something missing?

We have modified the sentence to clarify that high rates of hybridization combined with male-biased gene flow make guenons an ideal system to study how Haldane's rule affects patterns of introgression on sex-linked loci (ll. 79-84).

Line 76. Add reference to the statement.

We believe the reviewer refers to "... Y chromosomal and autosomal phylogenies generally agree in guenons, in line with the expectations under Haldane's rule...".

However, there is to our knowledge no direct reference for this statement beyond the logical expectation that no Y-chromosomal introgression occurs under Haldane's rule, leading to concordance between Y-chromosomal and autosome phylogenies, examples for which we cite earlier in the introduction (ll. 65-71).

Line 149 and beyond. In the text the pairwise nucleotide divergence is abbreviated as d_{xy} , but in some figures (Figs. 3C and S6) it is written in capital letters with XY as subscript. Please check for consistency throughout the manuscript.

We have updated this in the text and figure such that it is written as d_{XY} throughout, which is standard in the literature.

Line 151. I suggest adding "... Y chromosome must have been polymorphic in the ancestral population that gave rise to the mona and mitis/cephus lineages and remained polymorphic and unsorted along the branches of these groups."

We clarified this following the reviewer's suggestion.

Line 181. Add "autosomal" between "excess" and "allele".

Corrected accordingly.

Line 186. Throughout the manuscript you typically use "ref#" to cite work previously published, but here you used "Jensen et al. (2023)". Please check for consistency.

We changed this citation to have the format consistent throughout the manuscript.

Line 189. Add "likely" between "gene flow event" and "occurred", as that is an inference based on previous analysis.

We have modified the wording to clarify that the *cephus* group was the predominant donor of this gene flow event here following the reviewers suggestion (ln. 208). Since this statement is based on exhaustive analyses performed by us in a previous study, the predominant gene flow is well confirmed and not inferred here. However, it is still possible that low levels of gene flow occurred from the *mitis* group, as we acknowledge on ll. 260-261.

Line 197. I suggest adding "...in comparison to allele sharing between the later and *C. pogonias*..." to make the comparison clearer.

We removed the second part of the original sentence as it appeared to cause confusion (ln. 215). To clarify, we used *C. wolffi* and *C. pogonias* as comparisons, but we only quantified excess allele sharing between *C. denti* and *C. mitis*. In the original version it was possible to read it as if we compared the excess allele sharing between *C. denti* and *C. mitis* with that between *C. pogonias* and *C. mitis*, which is not the case.

Line 211 and beyond. Like the abbreviation of d_{xy} , there is inconsistency in the abbreviation of F_D , as it is sometimes abbreviated as f_D (e.g., line 581). Please check for consistency throughout the manuscript.

Thank you. Corrected throughout to f_d .

Line 213. It says, "...55 protein coding genes (Table S4)...", but the table only shows 54.

It should indeed be 54, but since the analyses was updated and substituted with topology-based inferences this comment is no longer relevant.

Line 235. I suggest using "support", instead of "confirm", as these are inferences.

Changed as suggested.

Line 335. It would be useful for the reader if it is more explicit what "genotyped sites" means in this sentence.

Clarified that we refer to sites in the reference genome that were retained following filtration.

Line 423. Provide the example and the reference of this "notable exception".

We agree that this was an unclear formulation, and have removed "notable exception". We were referring to the Y introgression that is found in the present study, that we return to in the next sentence. It should be easier to follow now (ll. 439-441).

Line 430. Change "showed" to "would have had", as you are only inferring this divergence.

Changed this according to the reviewers suggestion.

Line 435. I suggest using "strongly", instead of "unambiguously", as these are estimates based on simulations.

Changed this according to the reviewers suggestion.

Line 458. I suggest mentioning that in the orangutan study the expression levels were similar, and later in the paragraph discussing the type of expression data that would be needed to address this possibility in guenons.

We now clarify that there was no apparent increase in the expression of *CDY* in the orangutan study (ln. 479), and that testis expression data would be needed to address this in guenons (ln. 489).

Line 483. Consider here the possibility that males were not produced (or were infertile) in the first generations of hybridization, as described in my initial comment.

We have included this possibility earlier in the discussion (ll. 450-452), and removed this specific statement from the paragraph (see response to the initial comment).

Line 491. Include between brackets the number of samples used in this study for each taxon.

Included accordingly.

Line 537. Please be more explicit as to what “converting the genotypes to a chromosome-wide alignment” means. Does it mean that you first aligned your sequences to the reference *Macaca* Y chromosome?

Indeed we used the short read alignments against the reference macaque Y chromosome, and just converted this to fasta format. This is now clarified in the text (ll. 562-564).

Line 559. The four individuals for which the TSPY gene was Sanger sequenced are not included in Table S1. Please either add them to the table and/or provide information of their precedence (wild, captive, location, permits, etc.).

These individuals are now included in Table S1.

Line 563. Please explain how you “extracted the corresponding regions” and, if relevant, provide the code.

We now explain more in detail that this was done by blasting the Sanger sequenced data against the reference genome, and then converting these regions to fasta sequences (ll. 589-592).

Line 593. Are these “consecutive windows” only those that showed signatures of introgression?

This analysis has been removed.

Line 677. For clarity, add “...with the least amount of Y chromosome missing data”.

Added this accordingly.

Reviewer #2 (Remarks to the Author):

(also sent as attachment, since the equation may not read easily here)

In this manuscript, Jensen et al. detect a Y chromosome introgression event between two deeply diverged guenon species, namely from *C. mitis* into *C. denti*. The authors first reconstruct the phylogenetic tree for 22 previously reported guenon genomes, plus two newly sequenced species, detecting a discrepancy between the autosomal and the Y chromosomal phylogenetic trees. This discrepancy is unlikely to be caused by incomplete lineage sorting (ILS), since the Y chromosome shows much lower divergence than autosomes. D-statistics suggest low levels of autosomal introgression from *mitis* into *denti*, implying that the Y chromosome introgressed at low frequency. The authors then perform extensive simulations with and without selection, concluding that drift alone is unlikely to have driven the introgressed Y chromosome to fixation. The authors pose two possible explanations for selection acting on the Y chromosome, namely adaptive introgression and meiotic drive.

The rationale and results of the paper are well explained and well-organized, and finding of Y chromosome introgression between unusually deeply diverged species is an interesting case study for the speciation/admixture genetics community (as exceptions to the rule often help us understand the rule itself!). We agree that the scenario favored by the authors is currently the most reasonable explanation. However, we have several suggestions that we hope will help improve the manuscript overall.

Major comments

1) Much of the argument for selection comes from simulations in msprime and SLiM. However, we note that they can be entirely replicated using analytical methods alone, with the advantage of removing stochasticity, providing direct estimations of very low probabilities, and linking the results directly to population genetic theory. While we

understand that researchers differ in their attraction to simulation-based logic versus analytical logic, consider adding and/or replacing the simulations for the ILS analysis and the selection analysis (Figure 6C) following the suggestions below. For 6B (see below), we think simulations are not necessary at all since the probability of fixation of a neutral mutation is very well-worked out and familiar to most population/evolutionary geneticists.

First, using standard coalescent theory, one can calculate the probability that the Y chromosomes of *wolfi* and *denti* do not coalesce before reaching the split time with *mitis* clade. Taking the demographic parameters from the model inferred using BPP-MSci (fig. 5):

$$\Pr(\text{no coalescence between } wolfi \text{ and } denti) = \exp(-t/(2N_e/4)) = \exp\left[-\frac{(-600,000)/(2 \times 200,000/4)}{1}\right] = 0.0025$$

Where $2N_e/4$ is the effective population size of the Y chromosome, t is the time in generations between the *wolfi*/*denti* split and the *denti*/*mitis* split. Furthermore, in order for *denti* and *mitis* to coalesce deep in time due to ILS, this probability is further reduced by 1/3, meaning that there is a less than 0.1% chance that the Y chromosomal discrepancy is due to ILS. This is probably why the authors do not observe any of their 1000 simulations grouping *denti* and *mitis* together in Fig. 6B when Y chromosomal migration is 0 (but the probability is not 0!).

Having established that ILS is unlikely, we can assume (in the absence of selection) that the initial Y chromosomal frequency equals the probability of fixation. This explains the one-to-one relationship between the migration rate and the probability of *denti*/*mitis* monophyly in Fig. 6B. The uncertainty around the median is simply due to the stochasticity of sampling from a binomial of size 1000, with probability of success equal to the initial frequency. More simulations would decrease the uncertainty around the mean, but the expectation remains the same. Thus, Fig. 6B is not necessary, since it's based on the well-known finding that the probability of fixation under drift is equivalent to its original frequency.

Finally, Fig. 6C can also be replicated without the need for performing simulations. The procedure is a bit more complex, and involves using binomial sampling with selection for a Wright-Fisher population together with phase-type theory to calculate the probability of fixation based on the initial frequency and the selection coefficient. The procedure is explained in section 4.4 and Fig. 5 in Hobolth et al. 2024 (<https://doi.org/10.1016/j.tpb.2024.03.001>). We attach accompanying R code to replicate Fig. 6C of the manuscript using this approach. Regardless of what you chose to do, if simulations remain a main source of inference, please provide the code for replicating the SLiM and/or msprime runs.

We thank the reviewers for these elaborate and helpful comments and suggestions to our analytical approach. We agree that the main findings shown in Figure 6B can be derived analytically, and have replaced it with a visual representation of the initial Y chromosome frequency under various autosomal migration rates provided strict male migration (as the Y chromosome migration rate will be twice the autosomal rate, we think it is useful to visualize this separately). However, we believe that the simulation approach estimating *C. denti*/*C. mitis* monophyly is still merited, as the theoretical prediction of fixation probability of the introgressing Y chromosome (i.e., the initial allele frequency) does not incorporate the possibility that the Y chromosome introgressed from an ancestral *mitis* lineage (i.e., event 1 in Figure 5) and reached fixation in *C. denti* and *C. m. opisthostictus* through ILS. While this probability can also be derived analytically, such calculations would become quite complex and we think that simulations are easier to grasp for the general reader. We therefore moved the original Figure 6B into supplementary material (Figure S11) as an additional confirmation of the processes under different scenarios.

The reviewer also points out that the probability that the Y chromosome discordance is due to deep ILS can be estimated to less than 0.1 % based on the inferred demography. While we are grateful for this input, we believe that the patterns of

nucleotide divergence, as presented in Figure 3, convincingly excludes the possibility of deep ILS.

Last, the reviewer points out that also Figure 6C can be replicated without simulations, to show the theoretical expectations rather than the simulated outcomes. We have used the attached R code to recreate Figure 6C, which generates an almost identical figure (expected but also reassuring). Although we see the benefit of estimating the actual probabilities, we believe that the simulation approach in SLiM is more accessible and easier to recreate for the majority of readers, and prefer to keep the simulations to illustrate the effects of selection in Figure 6C. One benefit of the simulation approach is that it allows for segregating variants to be reported and does not make assumptions of their fixation. The code to replicate these simulations is available at https://github.com/axeljen/denti_ychrom_scripts, which is also stated in the “Code availability” statement in the manuscript.

2) The authors have not explicitly characterized the pseudoautosomal region (PAR). This is the only recombining region between the X and the Y chromosomes, and it might show interesting patterns of divergence or mutation (see Bergman and Schierup 2022, <https://doi.org/10.1186/s13059-022-02784-x>). It is also unclear whether PAR has been included in the divergence calculations, which might bias some of the inferences. A Y-chromosome-wide plot showing the areas included in the analyses would be helpful to clarify this point; there is a coverage plot in the supplement, but it's a bit hard to interpret, since it's based on coverage normalized to the mean.

Indeed this is an important point. PAR was not included in our original analyses of the Y-chromosome, as it is not assembled on the macaque Y reference (see below). In addition, our genotype filtration steps on the Y chromosome (excluding sites where any male was heterozygous, and sites with more than twice the average Y chromosomal coverage) largely remove any PAR. However, following the reviewer's suggestion, we have now included a more thorough identification and analysis of the PAR (ll. 240-248, Figures S7-S8). We used a coverage-based approach to identify PAR, and found one region of ~2.36 Mb on the X chromosome that shows patterns consistent with a PAR (PAR1), whereas no such region was evident on the reference Y chromosome. The explanation for the PAR being absent on the Y is most likely purely technical (i.e. misassembly). While PAR2 may be present in the macaques, the current quality of the genome assembly may preclude us from identifying it. In T-2-T chromosomal assemblies, however, independently originated PAR2 were found only in humans and bonobos, (Makova et al. 2024), whereas a recent study of primate Y chromosomes reported an independent acquisition of PAR2 in a guenon *Chlorocebus sabeaus* (Zhou et al. 2023). Please note, however, that since we are mapping to the macaque reference and using the above-described filtering steps, the *Chl. sabeaus* specific PAR2 does not impact our results. Following PAR identification, we have expanded our analyses to include nucleotide divergences and gene flow estimates in PAR and non-PAR segments of the X chromosome (Figure S8). We show the expected patterns of PAR behaving like the autosomes and the non-PAR X chromosome for introgression, and showing greater divergence than autosomes and non-PAR X chromosome (Bergman and Schierup 2022).

3) The manuscript comes down strongly in favor of a selective hypothesis (e.g., “Our analyses unambiguously suggest...”). We agree that this is a likely possibility, but the language of certainty should be modified throughout (and the authors should clarify that they consider the meiotic drive possibility a subset of selection here; or perhaps refer to the combination of drive and adaptive introgression as ‘non-neutral processes’). Part of the uncertainty involves use of point estimates for divergence dates and allele frequencies. For example, the introgressed Y is treated as fixed, but this is based on only 5 males. Apparent fixation with a sample of 5 is also compatible with high, but not fixed, frequencies, which would change the simulation estimates. Similarly, the split times (8 million years ago for autosomal divergence, 1.9 mya for Y) are treated as fixed, but both likely have uncertainty estimates around them that affect how “necessary” selection is to the story. And of course, clearly corroborative functional evidence has not yet been obtained.

We fully agree with the points brought up by the reviewers and have modified the level of confidence in our language throughout the manuscript (e.g., abstract, ll. 396, 400-401), to better reflect the uncertainties in our inferences. Obtaining functional evidence in a wild non-model primate species is challenging and will remain out-of-reach the time being, as is the case in many wild systems.

Minor comments

In Fig. 3C, the title of the plot states “Nucleotide diversity on the Y chromosome”, but consider editing the title, since you also show the diversity on the autosomes alongside the Y chromosome.

We changed the title of Figure 3C to “Nucleotide divergence (d_{XY}) on the Y chromosome and autosomes”.

Can the distribution of sizes of the putatively introgressed regions on the autosomes be used to estimate gene flow dates here as well? It would be helpful to see if the autosomal dating corroborates the timing estimated for the Y.

Although this is an interesting idea, we think that the old age and scarcity of introgression makes it difficult/impossible to accurately identify introgressed haplotypes. Furthermore, we have replaced the previous fd/dxy window outlier approach to identify introgressed regions in the autosomes/X chromosome with a topology based approach (Figure S6), following a suggestion from reviewer #3. While this approach is likely less prone to false positives, it is likely that the genomic regions identified as introgressed contain a mixture of introgressed and ancestral haplotypes, making an age estimate difficult.

Reviewer #3 (Remarks to the Author):

The authors study 57 samples of guenon species. These sequenced reads were mapped against the rhesus macaque reference genome.

They find a discrepancy in the Y chromosome tree vs the species tree between *C. Denti* and *C. Mitis* subspecies. The divergence of the Y chromosomes is >6 millions years. This is very exciting find and to my knowledge the most divergent Y chromosome introgression that we know of. It provides a notable exception to Haldanes rule. The autosomal divergence is 8 million years and the Y chromosome divergence is 2 million years. The authors confirm this finding using of TSPY genes in four additional *C. denti* males.

The authors conduct simulations to show that this is highly unlikely under neutrality and conclude that there must be some selection going on.

The look for a plausible mechanism for this and detect non-synonymous differences on the Y chromosome in 2 genes (KDM5D and USP9Y) and find a large region (1.35 Mb) has higher copy number suggestive of gene duplications - from humans we know these are very common but we do not know the function of them yet.

Overall I enjoyed the paper and find the conclusions well supported. Good job!

I only have a few comments:

1. In the section beginning at line 144 the authors want to distinguish between ILS and introgression using the divergence between the autosomes and Y chromosomes. They use the divergence measure D_{xy} - could you please provide units for this (mutations per basepair I assume)? In addition, I think it could be really useful to convert the D_{xy} on the y axis values in Figure 3 to coalescence times in years to make it easier to compare to Figure 1

We have included the nucleotide divergence scaled to coalescence time in years as a second axis in Figure 3. Please note that this should be interpreted with caution, as this measures the coalescence time between these specific sequences (i.e. species divergence time + additional time to coalescence in the ancestral population, which depends on the population size). They are thus not accurate estimates of species divergence times, and not directly comparable to the split times in Figure 1 and Figure 5.

2. Dstatistics

The informative statistic here to test for gene flow between *C. mitis* and *C. denti* would be on the form: (C. Wolfi, C. Denti ; C. mitis, rhesus macaque)

There is only support for geneflow in some individuals and I found table S3, D-statistics very helpful. However I am confused about the columns BBAA, and f4-ratio. Why do we need this information? And if there was an F4 ratio test what f4-values are being divided? I couldn't find this in the supplement but perhaps I missed it.

We sorted table S3 to display these relevant trios first, and highlighted them in green. Furthermore, we removed the BBAA and f4-ratio columns from this table, as they are not relevant to our result (they are included in the default output of Dsuite, which is why they were initially included in this table).

3. The Fd statistic and in combinations with dxy and 4 different filters seems very ad-hoc to me. How accurate is this method - how does it work with the data you have? If the authors want to use this to detect introgressed regions they should show how well it works for instance the accuracy and false positive rate of this statistic using simulated data under their inferred demography. Alternatively this section can be taken out of the paper.

We agree with the reviewer that while plausible, the combination of test statistics to arrive at introgressed loci is not well established. Therefore, we have replaced the fd/dxy outlier-based analyses of autosomal/X-chromosomal introgression, with a topology based approach (II. 229-238), as explained in the response to point 2 brought up by Reviewer 1. We considered removing this analysis entirely, as suggested by the reviewer. However, we believe that given the reported gene flow, any autosomal genes that have introgressed in the process may be of interest and hence the investigations will be incomplete without their inclusion.

Minor comments

Figure 1 - Can the authors please provide bootstrap values on the tree or some measure of variance. They write in the text that the grouping of *C. mitis* and *C. albogularis* is poorly resolved so that should be indicated in Figure1.

We have added support values (LPP) on the Astral tree in Figure 1A. Regarding the *C. mitis* and *C. albogularis*, however, these lineages are *taxonomically* unresolved - i.e. the support of their branches in the tree is complete, but they are not monophyletic species (as shown in Figure 1).

Fig 1 vs Fig S1

You use salmon in Fig S1 instead of red - but the colors look the same on my computer. Maybe just a typo

Changed this to be consistent.

line 234 and line 615 - needs units for the mutation rate

Added unit to mutation rate statements.

line 432 what is the unit of divergence? 10.8% divergence between human and chimpanzee (ref 39) is for mitochondrial! Nuclear divergence is more like 1-2% I believe.

Indeed all of these estimates are mitochondrial divergence, we clarified this in the text (ll. 444-450).

Reviewer #4 (Remarks to the Author):

REVIEWER COMMENTS

In the following, we respond to the comments provided by Reviewer 2 and 3, as other reviewers did not have additional comments on the revised version of the manuscript

Reviewer #2 (Remarks to the Author):

This is the second time we have seen this manuscript. As before, we continue to find the results interesting, and the scenario favored by the authors (introgression of a deeply diverged Y chromosome, followed by selection or meiotic drive to drive it to high frequency/fixation) plausible.

The revisions in this version improve the manuscript overall, although most of the changes are only clarificatory in nature. Engaging with the additional analyses or alternative explanations suggested by the reviewers could have strengthened the manuscript even further (e.g., alternative mechanisms that give rise to Haldane's Rule; the possibility that the results are not an exception to Haldane's Rule as typically interpreted—which the authors agree with in the rebuttal, but still appears in the title; analytical—as opposed to only simulation-based—support for the selection scenario).

We thank the reviewer for evaluating our revised version of the manuscript, and are pleased that they find it improved. The reviewer lists a number of suggestions for us to elaborate on further in the comment above, which we have considered point by point and give responses to below:

- 1. Alternative mechanisms that give rise to Haldane's Rule.**

We agree and acknowledge that there are additional mechanisms that may contribute to the reduced fitness often observed in heterogametic hybrids. We clarified this in the introduction (ll. 64-65). However, the mechanism behind Haldane's rule is merely a discussion point of this manuscript, as we are not able to explicitly test different mechanisms with the data at hand. Therefore, we believe that this topic is sufficiently covered with the current update.

- 2. The possibility that the results are not an exception to Haldane's Rule as typically interpreted (which the authors agree with in the rebuttal, but still appears in the title).**

As the reviewer points out, and as discussed in the first round of reviews, the typical interpretation of Haldane's rule is the lack of fertile, heterogametic F1 hybrids. Although a fertile male *C. denti* x *C. mitis* F1 hybrid is the simplest explanation to the Y chromosome introgression described in our manuscript, another possibility is that the Y chromosome introgressed via an already admixed population. This scenario would thus not necessarily require a fertile male F1 hybrid. We have included this possibility in the Discussion section following the first round of revisions and have now expanded on it, dedicating a full paragraph to this alternative interpretation (ll. 453-464).

In the current version of the manuscript, we refrain from referring to the Y chromosome introgression in *C. denti* explicitly as a direct exception to Haldane's rule. Rather, we discuss it as an exceptional event in the context of the scarcity of sex-limited chromosome introgression in the literature, which is likely caused predominantly by reduced fitness of heterogametic hybrids (i.e. indirectly caused by Haldane's rule). Therefore, we think that the title is justified, and serves the purpose of highlighting this unique event. Nevertheless, we are open to removing "Breaking the rule" from the title, pending editorial advice.

3. Analytical, as opposed to only simulation-based, support for the selection scenario.

We are thankful for the detailed descriptions in the previous review round on how to estimate the expected fixation probabilities. As we stated in the previous response, we ran the code provided by the reviewer and retrieved results indistinguishable from our simulation-based estimates. Therefore, we think that our results and conclusions are well supported by the current simulation-based tests. As the reviewers acknowledged in a previous comment, researchers have different preferences for analytical derivations versus simulations. . Furthermore, our simulations incorporate the possibility that the Y chromosome introgressed from an ancestral mitis lineage (i.e., event 1 in Figure 5) and reached fixation in *C. denti* and *C. m. opisthostictus* through ILS. We believe that simulations of such complex scenarios are easier to grasp than analytical derivations of the process.

We agree, however, that the results presented in the former Figure 6B were redundant, considering the simple and well-established relationship (fixation probability = initial allele frequency). We have therefore removed this panel from the main Figure 6 altogether, and refer to the theoretical expectations in the text (ll. 312-315).

One key addition is the attention to the PAR region of the X. Here, the authors now report that the PAR shows greater divergence between *C. mitis* and *C. denti* than on the rest of the X or the autosomes. While this result is consistent with patterns reported in the great apes in the absence of Y chromosome introgression, it is somewhat puzzling given that this region specifically is expected to recombine with the putatively introgressed mitis Y. In the scenario proposed by the authors, shouldn't the region therefore exhibit reduced divergence between *C. mitis* and *C. denti* (in the absence of selection against the introgressed mitis DNA)? To put this result in context, the authors could also have calculated the divergence between *C. denti* and *C. wolfi* on the X PAR and compared it with the divergence between *C. denti* and *C. mitis* (as shown in Figure S8). The *denti/wolfi* comparison would serve as the "baseline" case (i.e., no introgression but the same evolutionary history of divergence as *denti/mitis*).

We understand that our finding of lack of introgression in the PAR, despite Y chromosome introgression, may seem counterintuitive at first sight. Following the comment from the reviewer, we therefore expanded on this in the manuscript (ll. 246-253), and ran additional simulations to explore the expected linkage between an introgressing Y chromosome and PAR. In summary, we find that the PAR behaves similar

to the autosomes in terms of introgression rates, which we argue is expected for a number of reasons:

1. The recombination rate is much higher in the pseudo-autosomal region compared to the rest of the genome, due to the obligate crossover during male meiosis (up to 20 times higher recombination rates in PAR than on autosomes reported in humans, (Helena Mangs and Morris 2007; Hinch et al. 2014)). Therefore, the linkage will be weaker between the PAR and the Y compared to other intra-chromosomal loci.
2. Any male migrant from *C. mitis* to *C. denti* will contribute with twice the migration rate on the Y chromosome compared to the PAR, since the PAR follows an autosomal inheritance pattern.
3. Considering that the Y chromosome is haploid, whereas the PAR is diploid, the introgressing PAR can remain segregating even after the Y chromosome is fixed. Thus, purging of *C. mitis* PAR introgression may have continued after the fixation of the introgressing Y chromosome.
4. The X chromosome is expected to be more resistant to gene flow relative to the autosomes due to the lower recombination rate on the X and exposure of recessive alleles in the hemizygous sex (Fraïsse and Sachdeva 2021). This may accelerate purging of *C. mitis* introgression in the PAR relative to the autosomes.

Our simulations demonstrate that point 1 alone, i.e. the increased recombination rate in the PAR, is sufficient to break down the linkage between an introgressing Y and PAR (ll. 248-250, Figure S8D, also included below for ease of viewing). We found that the PAR is expected to show levels of introgressions similar to the autosomes already at a tenfold increase in recombination rate relative to the genome-wide average. Points 2-4 are not considered in these simulations, and are expected to further decrease the introgression rates in the PAR. We therefore think that our finding of indistinguishable rates of introgression in the PAR and the autosomes is expected.

Figure S8D. Simulations exploring the expected allele sharing (f_d) on the PAR in the event of Y chromosome introgression, with different PAR recombination rates. The “Y” locus was simulated as a non-recombining region of 50 kb, linked to a “PAR” region of 150 kb with either the same ($r\text{-factor} = 1$) or increased ($r\text{-factor} = 10$ and 20) recombination rate relative to the “Autosomes”, which were simulated as 50 kb regions separated by 1 bp with a recombination probability of 0.5, effectively reflecting unlinked loci. f_d was estimated in 50 kb loci along each replicate “genome” simulated (illustrated by gray connectors). Colored labels indicate mean, window-specific f_d across all replicates.

We have also clarified the pattern of increased divergence in the PAR relative to the non-PAR X and autosomes (ll. 250-253). In the absence of PAR introgression, this is an expected consequence of an increased substitution rate in the PAR relative to the rest of the genome (Bergman and Schierup 2022). A greater divergence in the PAR was found both between *C. denti* and *C. mitis*, and *C. denti* and *C. wolffi*. Both of these comparisons are shown in Figure S8, and we clarified this by referencing specific panels in this figure (ll. 250-253)

Minor comment:

- The response to reviewers indicated that Figure 6B, which shows the relationship between initial allele frequency and fixation under neutrality, had been moved to the supplement as Figure S11, but it remains in the main text of the revised manuscript that we saw. As noted in the initial review, this is a trivial result since this relationship is long-established by theory. In addition, the exact relationship between autosomal migration and initial Y chromosome frequency assumes that all migrants are males, which is not particularly well-justified.

We agree with the reviewer and have removed the figure, as stated above. We now only refer to the theoretical fixation probability in the text (ll. 312-315), and keep the Y chromosome simulation figure (which accounts for additional gene flow events and ILS) in the supplementary material (Figure S11).

Concerning our assumption that all migrants are male: We make it to be as conservative as possible in our estimates. If only a fraction of migrants are male, the initial frequency of the Y chromosome will be even lower than presented, making the achievement of high frequency/fixation even more exceptional, which would require stronger selection. The assumption of all-male migration is at least partially justified given predominant male-biased dispersal in guenons, as detailed on ll. 83-84, 414-415.

- It would be useful for the reader to refer to relevant supplementary figures in the methods section (such as referring to Figure 7B in the paragraph starting in line 614).

We have added references to supplementary material throughout the methods, as suggested by the reviewer (e.g. ll. 534, 594, 599, 625 etc.).

- In line 313, consider providing the probability for Y chromosomal introgression + ILS using analytical derivations.

We clarified the analytical expectations of Y chromosome fixation, given various migration rates from *C. mitis* to *C. denti* in the previous revision (ll. 312-315). However, we prefer the simulation-based approach to explore the probability of Y chromosome introgression through a combination of more ancestral introgression and ILS, as these analytical derivations would be very complex (i.e., the probability of Y chromosome introgression from the *mitis* ancestor into the eastern *mona* clade ancestor, followed by ILS in both lineages and subsequent fixation of the same allele in *C. mitis opisthostictus* and *C. denti*).

Reviewer #2 (Remarks on code availability):

The instructions on how to run the code are clear and straightforward. The code can be run smoothly, and results are reproducible.

Reviewer #3 (Remarks to the Author):

The authors have addressed all my comments. The manuscript is in great shape and I will be happy to see it published! Well done to the authors!

We are grateful to the reviewer for a thorough and constructive assessment of our work, which has contributed to improving the manuscript.

- My last comment (and this is just a suggestion) would be to change Figure 4 slightly. My issue is that the authors are performing a f4 test (or D-statistic) with rhesus macaque as outgroup. However in the Figure they only show the 3 populations that change and it appears they are doing a f3 test at first glance. I would recommend adding the P4 population to the facets in the figure or at least highlight it in the figure text.

We clarified this by adding *M. mulatta* as the outgroup in the panel headers (Figure 4).

Reviewer #3 (Remarks on code availability):

Very well documented github repository!

Comments for “Breaking the rule: An exceptional Y chromosome introgression between deeply divergent primate species” by Jensen et al.

In this manuscript, Jensen et al. detect a Y chromosome introgression event between two deeply diverged guenon species, namely from *C. mitis* into *C. denti*. The authors first reconstruct the phylogenetic tree for 22 previously reported guenon genomes, plus two newly sequenced species, detecting a discrepancy between the autosomal and the Y chromosomal phylogenetic trees. This discrepancy is unlikely to be caused by incomplete lineage sorting (ILS), since the Y chromosome shows much lower divergence than autosomes. D-statistics suggest low levels of autosomal introgression from *mitis* into *denti*, implying that the Y chromosome introgressed at low frequency. The authors then perform extensive simulations with and without selection, concluding that drift alone is unlikely to have driven the introgressed Y chromosome to fixation. The authors pose two possible explanations for selection acting on the Y chromosome, namely adaptive introgression and meiotic drive.

The rationale and results of the paper are well explained and well-organized, and finding of Y chromosome introgression between unusually deeply diverged species is an interesting case study for the speciation/admixture genetics community (as exceptions to the rule often help us understand the rule itself!). We agree that the scenario favored by the authors is currently the most reasonable explanation. However, we have several suggestions that we hope will help improve the manuscript overall.

Major comments

1) Much of the argument for selection comes from simulations in msprime and SLiM. However, we note that they can be entirely replicated using analytical methods alone, with the advantage of removing stochasticity, providing direct estimations of very low probabilities, and linking the results directly to population genetic theory. While we understand that researchers differ in their attraction to simulation-based logic versus analytical logic, consider adding and/or replacing the simulations for the ILS analysis and the selection analysis (Figure 6C) following the suggestions below. For 6B (see below), we think simulations are not necessary at all since the probability of fixation of a neutral mutation is very well-worked out and familiar to most population/evolutionary geneticists.

First, using standard coalescent theory, one can calculate the probability that the Y chromosomes of *wolfi* and *denti* do not coalesce before reaching the split time with *mitis* clade. Taking the demographic parameters from the model inferred using BPP-MSci (fig. 5):

$$\begin{aligned} \Pr(\text{no coalescence between } wolfi \text{ and } denti) &= \exp\left(-\frac{t}{2N_e/4}\right) \\ &= \exp\left(-\frac{600,000}{2 \times 200,000/4}\right) = 0.0025 \end{aligned}$$

Where $2N_e/4$ is the effective population size of the Y chromosome, t is the time in generations between the *wolfi/denti* split and the *denti/mitis* split. Furthermore, in order for *denti* and *mitis*

to coalesce deep in time due to ILS, this probability is further reduced by 1/3, meaning that there is a less than 0.1% chance that the Y chromosomal discrepancy is due to ILS. This is probably why the authors do not observe any of their 1000 simulations grouping *denti* and *mitis* together in Fig. 6B when Y chromosomal migration is 0 (but the probability is not 0!).

Having established that ILS is unlikely, we can assume (in the absence of selection) that the initial Y chromosomal frequency equals the probability of fixation. This explains the one-to-one relationship between the migration rate and the probability of *denti/mitis* monophyly in Fig. 6B. The uncertainty around the median is simply due to the stochasticity of sampling from a binomial of size 1000, with probability of success equal to the initial frequency. More simulations would decrease the uncertainty around the mean, but the expectation remains the same. Thus, Fig. 6B is not necessary, since it's based on the well-known finding that the probability of fixation under drift is equivalent to its original frequency.

Finally, Fig. 6C can also be replicated without the need for performing simulations. The procedure is a bit more complex, and involves using binomial sampling with selection for a Wright-Fisher population together with phase-type theory to calculate the probability of fixation based on the initial frequency and the selection coefficient. The procedure is explained in section 4.4 and Fig. 5 in Hobolth et al. 2024 (<https://doi.org/10.1016/j.tpb.2024.03.001>). We attach accompanying R code to replicate Fig. 6C of the manuscript using this approach. Regardless of what you chose to do, if simulations remain a main source of inference, please provide the code for replicating the SLiM and/or msprime runs.

2) The authors have not explicitly characterized the pseudoautosomal region (PAR). This is the only recombining region between the X and the Y chromosomes, and it might show interesting patterns of divergence or mutation (see Bergman and Schierup 2022, <https://doi.org/10.1186/s13059-022-02784-x>). It is also unclear whether PAR has been included in the divergence calculations, which might bias some of the inferences. A Y-chromosome-wide plot showing the areas included in the analyses would be helpful to clarify this point; there is a coverage plot in the supplement, but it's a bit hard to interpret, since it's based on coverage normalized to the mean.

3) The manuscript comes down strongly in favor of a selective hypothesis (e.g., "Our analyses unambiguously suggest..."). We agree that this is a likely possibility, but the language of certainty should be modified throughout (and the authors should clarify that they consider the meiotic drive possibility a subset of selection here; or perhaps refer to the combination of drive and adaptive introgression as 'non-neutral processes'). Part of the uncertainty involves use of point estimates for divergence dates and allele frequencies. For example, the introgressed Y is treated as fixed, but this is based on only 5 males. Apparent fixation with a sample of 5 is also compatible with high, but not fixed, frequencies, which would change the simulation estimates. Similarly, the split times (8 million years ago for autosomal divergence, 1.9 mya for Y) are treated as fixed, but both likely have uncertainty estimates around them that affect how "necessary" selection is to the story. And of course, clearly corroborative functional evidence has not yet been obtained.

Minor comments

- In Fig. 3C, the title of the plot states “Nucleotide diversity on the Y chromosome”, but consider editing the title, since you also show the diversity on the autosomes alongside the Y chromosome.
- Can the distribution of sizes of the putatively introgressed regions on the autosomes be used to estimate gene flow dates here as well? It would be helpful to see if the autosomal dating corroborates the timing estimated for the Y.

```

---
title: "Figure 6"
format: html
editor: visual
---

```{r setup}

Load packages
library(tidyverse)
library(ggthemes)
library(glue)
library(ggribes)

...

Fig. 6C

```{r}

fct = 100
# Population size
pop.sz <- round(200000/fct)
# Initial frequency of mutation
freq <- c(0.1, 0.2, 0.4, 0.6, 0.8, 1)/100
# Selection coefficients
# selec <- c(-0.1, 0, 0.05, 0.1, 0.2, 0.5)
selec <- c(0, 0.00001, 0.0001, 0.001, 0.01)*fct/2
res_2 <- tibble()
for (init_freq in freq){
  print(init_freq)
  for (j in 1:length(selec)) {
    # Save selection coefficient
    s <- selec[j]
    # Create empty sub-intensity matrix
    subprb.matrix <- matrix( 0, nrow=(pop.sz-1), ncol=(pop.sz-1))
    # Calculate vector of probabilities for success for the binomial
sampling
    prob_vec <- (1:(pop.sz-1))*(1+s)/((1:(pop.sz-1))*(1+s)+pop.sz-
(1:(pop.sz-1)))
    # For each frequency of the selected allele
    for (i in 1:(pop.sz-1)){
      # Save probability of success
      prob <- prob_vec[i]
      # Binomial sampling
      subprb.matrix[i,] <- dbinom( x=(1:(pop.sz-1)), size=pop.sz,
prob=prob)
    }
    # Compute exit rate vector for loss
    t1 <- dbinom(x=0, size=pop.sz, prob=prob_vec)
    # Compute exit rate vector for fixation
    # t2 <- dbinom(x=pop.sz, size=pop.sz, prob=prob_vec)
    # Compute initial probability vector
    initial.prb <- rep(0, (pop.sz-1))
    initial.prb[round(pop.sz*init_freq)] <- 1
  }
}

```

```

res_2 <- bind_rows(
  res_2,
  tibble(
    # Probability of loss
    t_loss_norm =
      (initial.prb%%*%
       Matrix::solve(diag(rep(1,length(initial.prb)))-subprb.matrix)%*%
       t1)[,1],
    # Probability of fixation
    t_fix_norm = 1-t_loss_norm,
    # Selection coefficient
    s = s/fct*2,
    init_freq = init_freq*100
  )
)
}
}
...

```{r}

res_2 |>
 pivot_longer(-c(s, init_freq)) |>
 ggplot() +
 geom_col(aes(as.character(init_freq), value, fill = name)) +
 geom_text(aes(as.character(init_freq), 0.5, label =
round(t_fix_norm*100)), data = res_2) +
 facet_wrap(~s, ncol = 1) +
 theme_few() +
 scale_fill_manual(values = c('lightblue', 'pink'))

ggsave(glue("expectation_plot_{fct}.pdf"), height = 7, width = 7)
...

Fig. 6B

```{r}

nsims <- 1000
x <- seq(0, 2, 0.1)/100
x_dens <- seq(0, 3, 0.1)/100
tab_density <- tibble()
for (i in x) {
  tab_density <- bind_rows(
    tab_density,
    tibble(
      x_plt = x_dens*100,
      y_plt = dbinom(x_dens*nsims, nsims, i),
      z_plt = i*100
    )
  )
}

tab_quantiles <- tibble()
for (i in x) {

```

```

tab_quantiles <- bind_rows(
  tab_quantiles,
  tibble(
    q05 = qbinom(0.05, nsims, i)/nsims*100,
    q25 = qbinom(0.25, nsims, i)/nsims*100,
    q50 = qbinom(0.5, nsims, i)/nsims*100,
    q75 = qbinom(0.75, nsims, i)/nsims*100,
    q95 = qbinom(0.95, nsims, i)/nsims*100,
    x = i*100
  ))
}

tab_density |>
  group_by(z_plt) |>
  mutate(
    y_plt = y_plt/max(y_plt)
  ) |>
  ggplot() +
  geom_tile(aes(z_plt, x_plt, fill = y_plt, color = y_plt)) +
  geom_vline(aes(xintercept = x),
             color = "white",
             data = tibble(x = x*100-0.05)) +
  scale_fill_viridis_c() +
  scale_color_viridis_c() +
  scale_x_continuous(expand = c(0, 0), breaks = x*100) +
  scale_y_continuous(expand = c(0, 0)) +
  labs(
    x = "Y-chromosomal migration C. mitis -> C. denti (%)",
    y = "Normalized binomial density"
  ) +
  geom_segment(aes(x = x-0.02, xend = x+0.02, y = q50, yend = q50),
              linewidth = 1,
              data = tab_quantiles) +
  geom_rect(aes(xmin = x-0.02, xmax = x+0.02, ymin = q25, ymax = q75),
           color = "black", fill = NA,
           data = tab_quantiles) +
  geom_segment(aes(x = x, xend = x, y = q05, yend = q95), data =
tab_quantiles)
...

```